# No Pose Estimation? No Problem: Pose-Agnostic and Instance-Aware Test-Time Adaptation for Monocular Depth Estimation

## Abstract

Monocular depth estimation (MDE), inferring pixel-level depths in single RGB images from a monocular camera, plays a crucial and pivotal role in a variety of AI applications demanding a three-dimensional (3D) topographical scene. In the real-world scenarios, MDE models often need to be deployed in environments with different conditions from those for training. Test-time (domain) adaptation (TTA) is one of the compelling and practical approaches to address the issue. Although there have been notable advancements in TTA for MDE, particularly in a self-supervised manner, existing methods are still ineffective and problematic when applied to diverse and dynamic environments. To break through this challenge, we propose a novel and high-performing TTA framework for MDE, named PITTA. Our approach incorporates two key innovative strategies: (i) pose-agnostic TTA paradigm for MDE and (ii) instance-aware image masking. Specifically, PITTA enables highly effective TTA on a pretrained MDE network in a pose-agnostic manner without resorting to any camera pose information. Besides, our instance-aware masking strategy extracts instance-wise masks for dynamic objects (e.g., vehicles, pedestrians, etc.) from a segmentation mask produced by a pretrained panoptic segmentation network, by removing static objects including background components. These masks serve as informative and useful cues for MDE during TTA and are used to selectively mask the depth map (i.e., output of the MDE network). To further boost performance, we also present a simple yet effective edge extraction methodology for the input image (i.e., a single monocular image) and depth map. Based upon these strategies, we develop a powerful TTA strategy for the MDE network by introducing and balancing two customized loss functions, namely, depth-refining loss and edge-guided loss. Extensive experimental evaluations on DrivingStereo and Waymo datasets with varying environmental conditions demonstrate that our proposed framework, PITTA, surpasses the existing state-of-the-art techniques with remarkable performance improvements in MDE during TTA. Code is provided as supplementary material.

## 1 Introduction

Monocular depth estimation (MDE) is a computer vision task that predicts the depth of each pixel in a single RGB image from a monocular camera Rajpal et al. (2023).[1] As a key technology for three-dimensional (3D) perception, MDE is envisioned to play a paramount and essential role in numerous AI applications such as autonomous driving, robotics, augmented reality, scene understanding, etc. In these applications, MDE models or networks are often required to be deployed in dynamic environments under diverse conditions—even rather different from those during training—where distribution or domain of real data varies continually Li et al. (2023); Wang et al. (2022), e.g., due to variations in lighting, weather, objects, etc. Test-time adaptation (TTA) is a compelling and effective solution to cope with this issue by enabling MDE networks to adapt to new, unseen domains or environments during inference with neither retraining nor access to source datasets used for training Kuznietsov et al. (2021); Li et al. (2023).

---

[1]Such an image will be referred to simply as a single monocular image throughout this paper whenever there is no ambiguity.

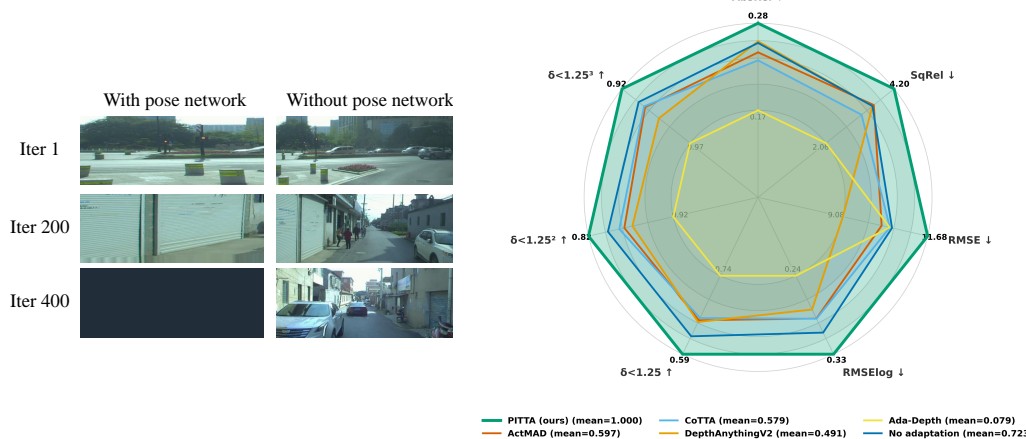

Figure 1: (a) Reconstructed images during TTA based on SfM assumption by following Godard et al. (2019) with and without pose estimation network. (b) TTA performance of our proposed method and other competing methods over 7 MDE metrics averaged over 4 different test cases on DrivingStereo and Waymo datasets.

According to these demands, several TTA methods for MDE have been developed in the recent studies, such as CoMoDa Kuznietsov et al. (2021) and Ada-Depth Li et al. (2023). Other TTA approaches applicable to MDE scenarios have also been reported in the literature, such as ActMAD Mirza et al. (2023) and CoTTA Wang et al. (2022). Despite merits and efficacy of these existing methods, several major technical limitations still remain. One critical and practical concern is that the MDE approaches in Kuznietsov et al. (2021); Li et al. (2023) are resorting to the Structure-from-Motion (SfM) assumption Zhou et al. (2017), which is however is highly likely to be violated in dynamic environments with domain shifts—realistic TTA situations. In addition to this, due to the reliance on the SfM assumption, adapting MDE networks using the TTA methods in Kuznietsov et al. (2021); Li et al. (2023) should be accompanied by additional adaptation of a separate network called pose estimation network. Thus, the performance of adapting the MDE network is sensitive and prone to the adaptability of the pose estimation network, but proper or optimal adaptation of the pose estimation network in practical TTA setups is still nontrivial and difficult, to our knowledge. Indeed, adapting the pose estimation network by directly extending a naive strategy based on the SfM assumption could lead to improper adaptations of both MDE network and pose estimation network during TTA, even exhibiting performance inferior to that of no adaptation. This serious phenomenon is empirically demonstrated in Fig. 1(a) with an illustrative example and in more detail in Appendix A with visualization results. Besides, the performance of other TTA approaches in Mirza et al. (2023); Wang et al. (2022) is generally suboptimal as they are not specialized (and thus, not suited) for MDE tasks. Furthermore, in the approaches of Li et al. (2023); Mirza et al. (2023), it is implicitly assumed that metadata and statistics of source datasets are available, respectively, which however may not be valid in strict real-world TTA settings without any access to source datasets. Above all, the most significant limitation is that the existing approaches in Kuznietsov et al. (2021); Li et al. (2023); Mirza et al. (2023); Wang et al. (2022) do not fully utilize available information for MDE during TTA, thereby hindering the potential and possibility for further performance improvements.

To break through these limitations, we introduce a pose-agnostic, instance-aware TTA framework for MDE, named PITTA, which achieves notable performance improvements as demonstrated in Fig. 1(b). One of the key technical innovations in our framework is that we present a novel pose-agnostic TTA paradigm for MDE, which does not require the SfM assumption. Unlike the existing approaches in Kuznietsov et al. (2021); Li et al. (2023), our pose-agnostic TTA paradigm enables a pretrained MDE network to be adapted with no camera pose information at all, eliminating the need for the adaptation of the pose estimation network during the adaptation of the MDE network. Another technical innovation is that we also devise a novel instance-aware masking strategy, which substantially enhances the adaptability of the MDE network during TTA. In this strategy, useful and valuable information on dynamic object instances (including vehicles, pedestrians, etc.) in monocular images are effectively utilized for MDE during TTA by means of so-called instance-wise masks, rather than using camera pose information as in prior works Kuznietsov et al. (2021); Li et al. (2023). On top of these, our framework further improves the performance by utilizing detected object boundaries as additional information for MDE and by balancing it with the dynamic instance

information, respectively, through effective edge extraction mechanism and loss function design. Our key technical contributions and breakthroughs in this work include the followings:

- We propose PITTA, a novel pose-agnostic and instance-aware TTA framework for MDE, which enables high-performing and effective adaptation of a pretrained MDE network in a pose-agnostic manner without resorting to any camera pose information.

- We devise an innovative and effective instance-aware masking strategy that can substantially enhance the adaptability of the MDE network during TTA by exploiting instance-wise masks for dynamic objects.

- We present an effective edge extraction mechanism to utilize detected object boundaries as additional informative cues for further enhancing MDE during TTA.

- We introduce two customized loss functions, depth-refining loss and edge-guided loss. Balancing these two loss functions via selective update of parameters enables appropriate adaptation of the MDE network.

- Through extensive experimental validation on widely used datasets, DrivingStereo dataset Yang et al. (2019) and Waymo dataset Sun et al. (2020), under diverse environmental conditions, we empirically demonstrate that PITTA markedly surpasses the existing state-of-the-art (SOTA) techniques in various MDE performance metrics and substantially enhances the adaptability of diverse MDE networks.

## 2  RELATED WORKS

**Monocular Depth Estimation.** In the literature, MDE has been widely and intensively studied as it is a long-standing and challenging task in the areas of computer vision Arampatzakis et al. (2024). MDE techniques can be roughly divided into two categories depending on the availability of labels (i.e., ground-truth depth values) during training—(i) supervised approaches such as Adabins Bhat et al. (2021), multi-scale MDE network Eigen et al. (2014), and NewCRFs Yuan et al. (2022); and (ii) self-supervised approaches such as MonoDepth2 Godard et al. (2019), SGDepth Guizilini et al. (2020), HR-Depth Lyu et al. (2021), Lite-Mono Zhang et al. (2023), and MonoViT Zhao et al. (2022). More details of related works on MDE with comprehensive literature review can be found in Appendix B.1. The supervised MDE techniques in Bhat et al. (2021); Eigen et al. (2014); Yuan et al. (2022) have enough potential to predict pixel-level depths with higher accuracy than the self-supervised ones. Nevertheless, a universal and practical concern is that acquiring labeled datasets is generally costly and time-consuming as they should be collected exhaustively in practice using LIDAR, and/or depth-measuring equipment like RGB-D sensors Geiger et al. (2012); Brachmann & Rother (2022). To address this issue, in the self-supervised MDE approaches Godard et al. (2019); Guizilini et al. (2020); Lyu et al. (2021); Zhang et al. (2023); Zhao et al. (2022), the depth values are estimated without the ground-truth labels, only from monocular images over different frames by invoking the SfM assumption. Unfortunately, however, the existing MDE methods in Godard et al. (2019); Guizilini et al. (2020); Lyu et al. (2021); Zhang et al. (2023); Zhao et al. (2022) all assumed that the environments or domains remain unchanged during both training and inference, and thus, these methods may perform poorly in practical TTA scenarios.

**Test-Time Adaptation for Monocular Depth Estimation.** Research on TTA for MDE remains largely unexplored in the literature, to our knowledge, and only a few works Kuznietsov et al. (2021); Li et al. (2023) have touched the issue in a self-supervised manner. More details of these works are provided in Appendix B.2. However, the MDE approaches in Kuznietsov et al. (2021); Li et al. (2023) are based on extensions of the earlier self-supervised MDE methods—such as in Godard et al. (2019); Yuan et al. (2022)—developed for non-TTA scenarios. Unfortunately, therefore, these methods are still ineffective and problematic when applied to diverse and dynamic environments, primarily on account of the reliance on the SfM assumption and the difficulty in properly adapting the pose estimation network.

## 3  OUR APPROACH: PITTA

In this work, the MDE task is performed via an off-the-shelf MDE network—such as in Godard et al. (2019); Guizilini et al. (2020); Lyu et al. (2021); Zhang et al. (2023); Zhao et al. (2022)—that outputs

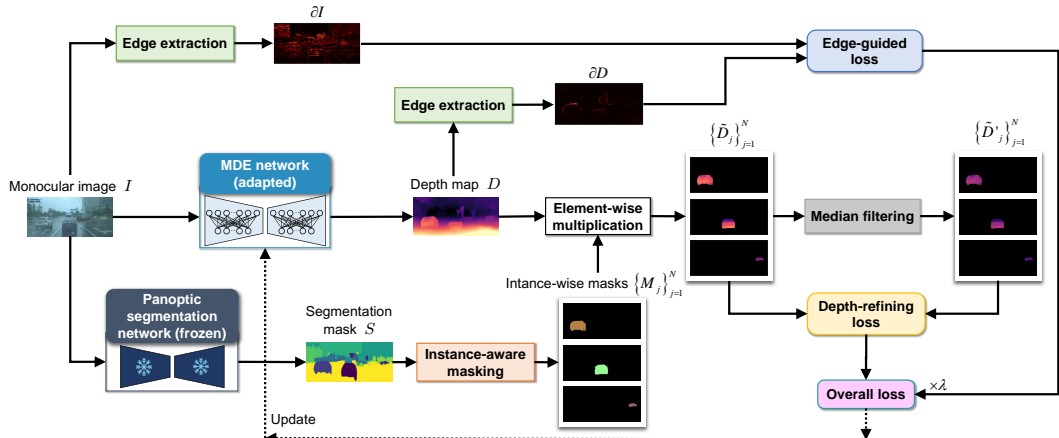

Figure 2: Overall architecture and schematic diagram of our TTA framework to adapt a pretrained MDE network given sequences of single RGB images from a monocular camera. Detailed computation procedures are presented in Algorithm 1 of Appendix C.

the pixel-wise depth estimates by taking a single monocular image as the input. Our goal is to adapt (or adjust) the MDE network pretrained on source (or training) datasets of monocular images to target (i.e., another) datasets—being in different domains or exhibiting distributional shifts from the source datasets—during the test (or inference) phase. We consider a realistic online situation where each sample (i.e., a single monocular image) in the target datasets is accessible sequentially, while the samples in the source datasets are not available at all. To accomplish this goal and break through the relevant technical challenges, we propose PITTA, a novel and high-performing TTA framework for MDE. The overall computation procedure of our framework is presented in Algorithm 1 of Appendix C and detailed in what follows.

## 3.1 OVERALL ARCHITECTURE

**Overview.** The overall architecture and schematic diagram of PITTA are depicted in Fig. 2. As can be seen from Fig. 2, our framework contains several modules. Firstly, there are two types of pretrained off-the-shelf networks: one is a pretrained MDE network (to be adapted) and the other is a pretrained panoptic segmentation network (frozen during the adaptation of the MDE network). On top of these networks, an additional innovative design, called instance-aware masking, is further employed in PITTA along with an effective edge extraction strategy for technical advancement and performance improvement of MDE during TTA. Further details of these modules are provided in the following.

**MDE Network.** At each time step, the MDE network predicts the depth of each pixel in the input image (i.e., a single RGB image from a monocular camera) denoted by $I \in \mathbb{R}^{H \times W \times C}$ in Fig. 2, where $H$, $W$, and $C$ denote height, width, and channel, respectively. The output of the MDE network is a collection of depth estimates for all the pixels, called depth map, denoted by $D \in \mathbb{R}^{H \times W}$ in Fig. 2. In PITTA, we primarily adopt a pretrained MDE network of MonoDepth2 in Godard et al. (2019), which is based on a U-Net architecture using ResNet-18 as the encoder. Further details of this network is provided in Appendix D.1. Also, in PITTA, we propose to adapt only parts—not the whole part—of the adopted MDE network to the test datasets during TTA. The reasons for this will be detailed later in Section 3.2. In general, another off-the-shelf MDE network such as in Guizilini et al. (2020); Lyu et al. (2021); Zhang et al. (2023); Zhao et al. (2022) can also be adopted in our framework.

**Pose-Agnostic TTA Paradigm.** The existing MDE approaches in Godard et al. (2019); Guizilini et al. (2020); Kuznietsov et al. (2021); Li et al. (2023); Lyu et al. (2021); Zhang et al. (2023); Zhao et al. (2022) relying on the SfM assumption are ineffective and problematic for TTA as they require the knowledge of camera pose information with continual adjustment, which is however rather challenging in practice. For more detailed exposition, let $(x, y)$ denote the two-dimensional (2D) coordinate of each pixel of the image image $I$ in a particular frame and $(x', y')$ denote the 2D

coordinate of the corresponding pixel in another frame. Also, let $D(x,y) > 0$ denote the depth value for pixel $(x,y)$, i.e., pixel intensity of $D$ at $(x,y)$. The common approach in the the existing methods is to retrieve the depth information from images in different frames based on the following 3D geometry relationship:

$$K \left( RK^{-1} \begin{bmatrix} x \\ y \\ 1 \end{bmatrix} D(x,y) + t \right) = z' \begin{bmatrix} x' \\ y' \\ 1 \end{bmatrix} \tag{1}$$

where $K \in \mathbb{R}^{3\times3}$ is a (known) camera intrinsic matrix and $z'$ corresponds to the last entry of the $3 \times 1$ vector in the left side of equation 1. Also, $R \in \mathbb{R}^{3\times3}$ and $t \in \mathbb{R}^{3\times1}$ represent rotation matrix and transition vector, respectively, which are typically unknown and predicted by a pretrained pose estimation network such as in Godard et al. (2019). Note that to make use of the relationship in equation 1 for TTA (i.e., adaptation of $D(x,y)$), it is required to adapt $R$ and $t$ accordingly as well. To this end, proper adaptation of the pose estimation network is essential, which is however nontrivial and rather challenging. To overcome this critical challenge effectively and uncomplicatedly, in PITTA, we present a new and efficient TTA paradigm for MDE in a camera pose-agnostic manner. The core idea of our TTA approach is to adapt the MDE network with no (prior) knowledge of camera pose information:

$$R = \begin{bmatrix} 1 & 0 & 0 \\ 0 & 1 & 0 \\ 0 & 0 & 1 \end{bmatrix}, \quad t = \begin{bmatrix} 0 \\ 0 \\ 0 \end{bmatrix}, \tag{2}$$

by simply presuming that there is no pixel transition across images in different frames. Our TTA approach is indeed useful and effective as it liquidates the need for acquisition of accurate camera pose estimates as well as reliance on a pose estimation network during the adaptation of the MDE network. Furthermore, it is worth noting that even without the exploitation of camera pose information, our framework markedly surpasses the existing techniques in a variety of TTA tasks for MDE, as will be demonstrated by the extensive and intensive experimental validations presented in Section 4.2.

**Panoptic Segmentation Network.** In PITTA, we also employ a pretrained panoptic segmentation network, the output of which, called panoptic segmentation mask, denoted by $S \in \mathbb{R}^{H\times W}$ in Fig. 2 will be used subsequently for the instance-aware masking of depth map $D$. We primarily adopt a network in Cheng et al. (2022), Mask2Former, which utilizes the tiny version of Swin Transformer Liu et al. (2021) as backbone. Further details of Mask2Former are provided in Appendix **?**. Another off-the-shelf panoptic segmentation network such as in Hu et al. (2023); Kim et al. (2020) can also be employed in PITTA. The panoptic segmentation network produces a panoptic segmentation mask for the input image, which represents pixel-wise semantic distinctions for individual instances of objects. Specifically, each pixel is assigned a pair of an (unique) instance identifier (ID) $i \in \mathcal{I}$ and the corresponding object (or semantic) label $\ell_i \in \mathcal{L}$, where $\mathcal{L}$ and $\mathcal{I}$ denote the sets of object labels (e.g., for indicating person, car, etc.) and instance IDs (e.g., for indicating multiple people, different cars, etc.), respectively.

**Instance-Aware Masking.** Another innovation in our framework is to incorporate an instance-aware masking strategy into TTA for MDE. Note that as described mathematically in equation 1, the key to MDE is to capture and extract the depth information from variations or changes in different frames, implying that dynamic objects (e.g., cars, people, animals, moving machinery, etc.) contain useful information for MDE. Unfortunately, however, most real-world monocular images include numerous static objects including the background elements, which even occupy overwhelmingly large spatial portions of the images. Obviously, the dominance of such static components significantly hinders the extraction of useful depth information and acquisition of accurate depth estimates (i.e., severely limits the MDE performance) during TTA. To address this critical issue, in our instance-aware masking strategy, we effectively suppress the impacts of the static objects on MDE during TTA by leveraging the so-called instance-wise masks. Meanwhile, it has been widely recognized that among various visual cues, identifying the information on objects' or instances' shapes plays the most important role in many computer vision tasks including MDE Landau et al. (1988). As a well-known example, Transformer-based networks tend to concentrate more on the shape information, which in turn contributes to their superior generalization behaviors to convolutional neural networks (CNNs) that tend to concentrate more on the texture information Ballester & Araujo (2016); Paul & Chen (2022). Inspired by these useful insights, in our instance-aware masking strategy, we produce

the instance-wise masks from the panoptic segmentation mask to effectively extract the useful information on each instance's shape and we further utilize these masks to improve depth predictions. We create the instance-wise masks only for the dynamic objects since the static ones are less informative for MDE during TTA as mentioned before.

Our instance-aware masking strategy proceeds as follows. Let $\mathcal{O} \subset \mathcal{L}$ denote the set of labels for the dynamic objects and $\mathcal{N} \subset \mathcal{I}$ denote the set of the corresponding instance IDs (i.e., the set of $i$'s corresponding to $\ell_i \in \mathcal{O}$) with cardinality $N$. Then $N$ different segmentation masks for $N$ distinct dynamic instances in $\mathcal{N}$ are produced, which are referred to as the instance-wise masks in our framework and are denoted by $M_j \in \mathbb{R}^{H \times W}$, $j = 1, \cdots, N$, in Fig. 2. Specifically, for such a segmentation mask corresponding to a dynamic instance $i \in \mathcal{N}$, the intensity of each pixel $(x, y)$ is computed as

$$g_i(x,y) = \begin{cases} i, & \text{if } (x,y) \in \mathcal{R}_i \\ 0, & \text{otherwise} \end{cases} \tag{3}$$

where $\mathcal{R}_i$ is the segment assigned to the instance-object pair $(i, \ell_i) \in \mathcal{N} \times \mathcal{O}$. Note that by equation 3, the shape information about all the instances irrelevant to $i \in \mathcal{N}$ (including all static objects/instances) can be removed while maintaining the shape information of a dynamic instance $i \in \mathcal{N}$ of interest. To this end and for more clear and delicate distinctions between the selected and remaining instances, in PITTA, the depth map is refined by applying the instance-wise masks as

$$\tilde{D}_j = D \odot M_j, \quad j = 1, \cdots, N, \tag{4}$$

where $\odot$ stands for the element-wise multiplication. These masked depth maps will be further used in the loss function design to properly refine the depth estimates.

**Edge Extraction.** We also discern that the boundaries (or edges) of both dynamic and static objects in images are highly useful and informative for MDE during TTA as they can offer additional structural or geometric cues that can further enhance or refine the depth predictions, for instance, by properly guiding the alignment of predicted depth discontinuities with true object boundaries Godard et al. (2017). Inspired by this, in PITTA, we introduce a weighted edge extraction approach based on the Laplacian approximation. In our edge extraction methodology, we construct the edge maps for the input image and the predicted depth map, denoted by $\partial I \in \mathbb{R}^{H \times W}$ and $\partial D \in \mathbb{R}^{H \times W}$ in Fig. 2, respectively, via

$$\partial I(x,y) = U(x,y) \times \big| \bar{I}(\min\{x+1, H-1\}, y) + \bar{I}(\max\{x-1, 0\}, y)$$
$$+ \bar{I}(x, \min\{y+1, W-1\}) + \bar{I}(x, \max\{y-1, 0\}) - 4\bar{I}(x,y) \big|, \tag{5}$$

$$\partial D(x,y) = V(x,y) \times \big| D(\min\{x+1, H-1\}, y) + D(\max\{x-1, 0\}, y))$$
$$+ D(x, \min\{y+1, W-1\} + D(x, \max\{y-1, 0\}) - 4D(x,y) \big|, \tag{6}$$

for $x \in \mathcal{H} \triangleq \{0, 1, \cdots, H-1\}$ and $y \in \mathcal{W} \triangleq \{0, 1, \cdots, W-1\}$, where $U(x,y)$ and $V(x,y)$ are nonnegative weights. Also, $\bar{I}(x,y) = \frac{1}{C} \sum_{z \in \mathcal{C}} I(x, y, z)$ for $x \in \mathcal{H}$ and $y \in \mathcal{W}$, where $\mathcal{C} \triangleq \{0, 1, \cdots, C-1\}$. The extracted edge maps in equation 19 and equation 6 will be exploited in the loss function design for more precise depth estimation and refinement.

## 3.2 TTA METHODOLOGY

**Loss Functions.** PITTA adopts the two loss functions for adaptation of the MDE network: one is the depth-refining loss $L_{\text{d}}$ and the other is the edge-guided loss $L_{\text{e}}$. In the depth-refining loss, we leverage the median filtering technique to denoise the masked depth maps $\{\tilde{D}_j\}_{j=1}^N$ while retaining edge details as follows:

$$\tilde{D}_j'(x,y) = \text{median}\Big\{ \tilde{D}_j(p,q) : p \in \big[\max\{x-r, 0\}, \min\{x+r, H-1\}\big],$$
$$q \in \big[\max\{y-r, 0\}, \min\{y+r, W-1\}\big] \Big\}, \ j = 1, \cdots, N, \tag{7}$$

for $x \in \mathcal{H}$ and $y \in \mathcal{W}$, where $r = \lfloor s/2 \rfloor$ and $s$ denotes the window size. The denoised versions $\{\tilde{D}_j'\}_{j=1}^N$ are used as pseudo labels for the depth estimates. The depth-refining loss is formulated as

the squared Euclidean distance between the masked depth maps and their denoised counterparts as follows:

$$L_{\mathrm{d}} = \frac{1}{N} \sum_{j=1}^{N} \left\| \mathrm{vec}(\tilde{D}_j) - \mathrm{vec}(\tilde{D}'_j) \right\|_1 = \frac{1}{N} \sum_{j=1}^{N} \sum_{(x,y) \in \mathcal{H} \times \mathcal{W}} \left| D_j(x,y) - D'_j(x,y) \right| \tag{8}$$

where $\mathrm{vec}(\cdot)$ is the vectorization operator. Meanwhile, to compensate for potential misalignment or discrepancy between the edge maps $\partial I$ and $\partial D$–corresponding to the input image and the predicted depth map, respectively–with additionally taken into consideration their inherent sparse nature, we also introduce the edge-guided loss, which is defined as the Manhattan distance between the two edge maps $\partial I$ and $\partial D$ as follows:

$$L_{\mathrm{e}} = \left\| \mathrm{vec}(\partial I) - \mathrm{vec}(\partial D) \right\|_1 = \sum_{(x,y) \in \mathcal{H} \times \mathcal{W}} \left| \partial I(x,y) - \partial D(x,y) \right|. \tag{9}$$

The above two loss functions $L_{\mathrm{d}}$ and $L_{\mathrm{e}}$ are then integrated into one overall loss function via a weighted combination:

$$L = L_{\mathrm{d}} + \lambda L_{\mathrm{e}} \tag{10}$$

where $\lambda \geq 0$ controls the tradeoff between the two loss functions $L_{\mathrm{d}}$ and $L_{\mathrm{e}}$.

**Adaptation of MDE Network.** To effectively cope with the catastrophic forgetting issue while maintaining the generalization ability during TTA, in our framework, we suggest to adapt only some parts of the MDE network while fixing the remaining parts. It should be noted that our approach of selectively adapting the MDE network is clearly distinct from and more general than prior approaches in Kuznietsov et al. (2021); Li et al. (2023) adapting the whole part of the MDE network. Specifically, let $\theta$ denote a set of selected parameters to be adapted in the MDE network. PITTA continually adapts $\theta$ by minimizing the overall loss function $L$ in equation 10 as follows:

$$\theta \leftarrow \theta - \alpha \nabla_\theta L \tag{11}$$

where $\alpha \geq 0$ is the learning rate and $\nabla_\theta$ denotes the gradient operator with respect to $\theta$.

## 4 EXPERIMENTS

### 4.1 EXPERIMENTAL SETUP

We examine the performance of our framework and compare it with other recently developed competing TTA techniques including CoMoDa Kuznietsov et al. (2021), Ada-Depth Li et al. (2023), ActMAD Mirza et al. (2023), and CoTTA Wang et al. (2022). For fair comparisons, these competing methods are adjusted to accommodate our depth-refining loss with instance-aware masking strategy by using the panoptic segmentation network. We also report the performance of DepthAnythingV2 Yang et al. (2024)—a synthetic data-based full training approach—as benchmark for our proposed method. In both our framework and other competing approaches, the MDE network of MonoDepth2 Godard et al. (2019) pretrained on the KITTI dataset Geiger et al. (2012)—a standard benchmark dataset for MDE—is adopted as backbone and it is adapted to two other different datasets during TTA: (i) DrivingStereo dataset Yang et al. (2019) and (ii) Waymo dataset Sun et al. (2020). In PITTA, we also employ Mask2Former Cheng et al. (2022) with Swin-Tiny Liu et al. (2021) backbone as the panoptic segmentation network. For test cases with MonoDepth2, the adapted parameters in PITTA are chosen as the parameters of batch normalization (BN) layers in the encoder of the MDE network, i.e., $\theta = \gamma \cup \beta$, where $\gamma$ and $\beta$ represent the sets of scale and shift parameters in the BN layers of the encoder, respectively. In the experimental evaluations, we measure the following popular performance metrics for MDE Eigen et al. (2014): absolute relative error (AbsRel), square relative error (SqRel), root mean square error (RMSE), log-scale RMSE (RMSElog), and threshold accuracies ($\delta < 1.25$, $\delta < 1.25^2$, and $\delta < 1.25^3$). More details about these performance metrics are provided in Appendix E. Further details on the experiment setting and dataset preprocessing can be found in Appendices F and G, respectively.

### 4.2 RESULTS AND DISCUSSION

Table 8 shows results of our framework and other approaches on the adaptation to the DrivingStereo dataset for test cases under diverse weather conditions—foggy, rainy, and mixture of all weather

Table 1: Results of our proposed method and other competing methods applied to adaptation of MDE network of MonoDepth2 pretrained on KITTI dataset to DrivingStereo dataset. In each test case, numbers in **bold** and underline indicate the best performance and the second best performance, respectively. Also, "All" indicates a test case concatenating cloudy, foggy, rainy, and sunny weather conditions in turn.

| Weather | Method | AbsRel ↓ | SqRel ↓ | RMSE ↓ | RMSElog ↓ | $\delta < 1.25$ ↑ | $\delta < 1.25^2$ ↑ | $\delta < 1.25^3$ ↑ |
|---|---|---|---|---|---|---|---|---|
| | No adaptation | 0.143 | 1.952 | 9.817 | 0.218 | 0.812 | 0.937 | 0.974 |
| | CoMoDA | 0.566 | 10.914 | 21.392 | 0.657 | 0.249 | 0.467 | 0.660 |
| | CoTTA | 0.175 | 2.344 | 9.899 | 0.243 | 0.750 | 0.920 | 0.969 |
| Foggy | ActMAD | 0.176 | 2.403 | 11.157 | 0.262 | 0.730 | 0.902 | 0.957 |
| | Ada-Depth | 0.208 | 3.306 | 9.884 | 0.227 | 0.719 | 0.884 | 0.948 |
| | DepthAnythingV2 | 0.164 | 2.741 | 12.891 | 0.278 | 0.736 | 0.880 | 0.944 |
| | **PITTA (ours)** | **0.127** | **1.579** | **8.600** | **0.195** | **0.840** | **0.951** | **0.980** |
| | No adaptation | 0.245 | 3.641 | 12.282 | 0.310 | 0.600 | 0.852 | 0.945 |
| | CoMoDA | 0.701 | 18.848 | 22.759 | 0.727 | 0.303 | 0.497 | 0.653 |
| | CoTTA | 0.225 | 3.127 | 11.057 | 0.286 | 0.638 | 0.875 | 0.960 |
| Rainy | ActMAD | 0.256 | 3.631 | 12.399 | 0.328 | 0.590 | 0.818 | 0.936 |
| | Ada-Depth | 0.322 | 5.623 | 10.782 | 0.387 | 0.572 | 0.794 | 0.897 |
| | DepthAnythingV2 | 0.208 | 3.105 | 12.480 | 0.293 | 0.657 | 0.860 | 0.946 |
| | **PITTA (ours)** | **0.195** | **2.532** | **10.315** | **0.254** | **0.685** | **0.905** | **0.973** |
| | No adaptation | 0.181 | 2.446 | 9.536 | 0.247 | 0.749 | 0.913 | 0.965 |
| | CoMoDA | 0.577 | 11.736 | 20.289 | 0.673 | 0.277 | 0.497 | 0.678 |
| | CoTTA | 0.200 | 2.619 | 9.657 | 0.265 | 0.710 | 0.899 | 0.961 |
| All | ActMAD | 0.172 | 2.190 | 9.543 | 0.244 | **0.758** | 0.913 | 0.967 |
| | Ada-Depth | 0.272 | 4.417 | 10.512 | 0.339 | 0.624 | 0.833 | 0.921 |
| | DetphAnythingV2 | 0.187 | 2.669 | 11.883 | 0.289 | 0.705 | 0.880 | 0.946 |
| | **PITTA (ours)** | **0.171** | **2.016** | **9.200** | **0.241** | **0.758** | **0.918** | **0.968** |

Table 2: Results of our proposed method and other competing methods applied to adaptation of MDE network of MonoDepth2 pretrained on KITTI dataset to Waymo dataset. Other conventions are the same as in Table 8.

| Method | AbsRel ↓ | SqRel ↓ | RMSE ↓ | RMSElog ↓ | $\delta < 1.25$ ↑ | $\delta < 1.25^2$ ↑ | $\delta < 1.25^3$ ↑ |
|---|---|---|---|---|---|---|---|
| No adaptation | 0.219 | 2.912 | 9.060 | 0.284 | 0.658 | 0.879 | 0.955 |
| CoMoDA | 0.543 | 8.098 | 14.309 | 0.597 | 0.316 | 0.517 | 0.693 |
| CoTTA | 0.273 | 4.239 | 10.347 | 0.331 | 0.584 | 0.835 | 0.933 |
| ActMAD | 0.230 | 2.559 | 8.866 | 0.292 | 0.621 | 0.873 | 0.960 |
| Ada-Depth | 0.315 | 3.462 | 9.809 | 0.377 | 0.451 | 0.771 | 0.921 |
| DetphAnythingV2 | 0.221 | 2.431 | 9.473 | 0.309 | 0.614 | 0.849 | 0.943 |
| **PITTA (ours)** | **0.199** | **2.097** | **8.206** | **0.266** | **0.670** | **0.896** | **0.967** |

conditions. It could be observed that for each test case, PITTA surpasses the other TTA methods in all performance metrics with remarkable improvements. This thus proves the superiority, effectiveness, and robustness of our TTA strategy for MDE based on pose-agnostic adaptation and instance-aware masking. Table 2 also presents results of the various methods on the adaptation to the Waymo Dataset—known as a more challenging dataset than the DrivingStereo dataset Yang et al. (2019). It shows that our framework attains the best performance in all test cases, thereby validating its universality and resilient adaptability across various domains. In Table 3, the performance of our framework is further demonstrated when it is applied to adaptations of other MDE networks—SGDepth Guizilini et al. (2020), HR-Depth Lyu et al. (2021), Lite-Mono Zhang et al. (2023), and MonoViT Zhao et al. (2022)—that are pretrained on the KITTI dataset to the DrivingStereo dataset. More results of our proposed method for these MDE networks can be found in Appendix H.1. Extended results for Mask2Former—the panoptic segmentation network adopted in PITTA—with different backbones are also available in Appendix H.2. It shows that PITTA can effectively adapt different types of MDE networks with performance improvements, where adapting SGDepth is observed to be the most effective by exhibiting the most significant improvements due to the network-architectural similarity to our framework. This manifests versatility of our proposed adaptation strategy.

### 4.3 ABLATION STUDY

**Effect of Loss Functions.** As an ablation study on our framework, the tradeoff between the depth-refining loss $L_d$ and edge-guided loss $L_e$ is examined in Table 4 in terms of the AbsRel metric and threshold accuracy with $\delta < 1.25$ for the case of adaptation to the Waymo dataset. The results demonstrate that the performance of PITTA is sensitive to the choice of the tradeoff parameter $\lambda$. Specifically, as $\lambda$ increases, the performance initially improves (up to $\lambda = 0.2$) and then deteriorates, indicating that there exists an optimal operating point with respect to $\lambda$. This in turn suggests that in practical design, more careful consideration is needed in the selection of $\lambda$ as PITTA will be most

Table 3: Results of our proposed method applied to adaptations of various MDE networks pretrained on KITTI dataset to DrivingStereo dataset. Other conventions are the same as in Table 8.

| MDE Net | Weather | AbsRel ↓ | SqRel ↓ | RMSE ↓ | RMSElog ↓ | $\delta < 1.25$ ↑ | $\delta < 1.25^2$ ↑ | $\delta < 1.25^3$ ↑ |
|---|---|---|---|---|---|---|---|---|
| SGDepth | No adaptation | 0.207 | 2.527 | 9.738 | 0.290 | 0.725 | 0.893 | 0.948 |
| | Sunny | 0.174 | 1.955 | 8.334 | 0.245 | 0.781 | 0.922 | 0.965 |
| | All | 0.170 | 1.809 | 8.305 | 0.244 | 0.784 | 0.922 | 0.964 |
| HR-Depth | No adaptation | 0.164 | 1.694 | 7.617 | 0.225 | 0.783 | 0.938 | 0.977 |
| | Sunny | 0.171 | 1.711 | 7.970 | 0.240 | 0.787 | 0.924 | 0.964 |
| | All | 0.169 | 1.865 | 9.022 | 0.255 | 0.776 | 0.913 | 0.959 |
| MonoViT | No adaptation | 0.150 | 1.609 | 7.648 | 0.211 | 0.815 | 0.943 | 0.979 |
| | Sunny | 0.141 | 1.425 | 7.528 | 0.207 | 0.826 | 0.948 | 0.979 |
| | All | 0.150 | 1.550 | 7.609 | 0.213 | 0.814 | 0.938 | 0.977 |
| Lite-Mono | No adaptation | 0.184 | 2.166 | 8.383 | 0.248 | 0.766 | 0.918 | 0.965 |
| | Sunny | 0.171 | 1.844 | 7.970 | 0.240 | 0.787 | 0.924 | 0.964 |
| | All | 0.173 | 2.079 | 9.406 | 0.243 | 0.750 | 0.915 | 0.968 |
| DetphAnythingV2 | No adaptation | 0.194 | 2.473 | 11.197 | 0.300 | 0.707 | 0.884 | 0.944 |
| | Sunny | 0.194 | 2.498 | 11.322 | 0.302 | 0.708 | 0.882 | 0.943 |
| | All | 0.198 | 3.001 | 12.593 | 0.305 | 0.684 | 0.857 | 0.935 |

Table 4: Ablation study on $\lambda$ regarding tradeoff between two loss functions $L_{\rm d}$ and $L_{\rm e}$ in terms of AbsRel and threshold accuracy $\delta < 1.25$ for adaptation to Waymo dataset. Other conventions are the same as in Table 8.

| $\lambda$ | 0 | 0.1 | 0.2 | 0.3 | 0.4 | 0.5 | 0.6 | 0.7 | 0.8 | 0.9 | 1.0 | $\infty$ |
|---|---|---|---|---|---|---|---|---|---|---|---|---|
| AbsRel ↓ | 0.236 | 0.203 | 1.999 | 0.200 | 0.202 | 0.204 | 0.205 | 0.206 | 0.207 | 0.208 | 0.208 | 0.218 |
| $\delta < 1.25$ ↑ | 0.618 | 0.644 | 0.670 | 0.668 | 0.663 | 0.657 | 0.654 | 0.652 | 0.649 | 0.647 | 0.647 | 0.627 |

Table 5: Ablation study on the number of adapted parameters in the MDE network of MonoDepth2 in terms of AbsRel and threshold accuracy $\delta < 1.25$ for adaptation to Waymo dataset. "CNN $a\%$ + BN $b\%$" indicates that only the parameters of last $a\%$ of CNN layers and last $b\%$ of BN layers in the encoder of the MDE network are adapted. Other conventions are the same as in Table 8.

| fraction | CNN 0% + BN 50% | CNN 0% + BN 100% | CNN 20% + BN 80% | CNN 40% + BN 60% | CNN 60% + BN 40% |
|---|---|---|---|---|---|
| AbsRel ↓ | 0.214 | 0.199 | 0.828 | 0.602 | 0.770 |
| $\delta < 1.25$ ↑ | 0.636 | 0.670 | 0.241 | 0.248 | 0.284 |

effective with modest values of $\lambda$. More results on this ablation study for other metrics are available in Appendix H.3.

**Impact of Number of Adapted Parameters.** In Table 5, we conduct another ablation study on the number of adapted parameters of the encoder (i.e., cardinality of $\theta$) in the MDE network of MonoDepth2 based on AbsRel metric and threshold accuracy with $\delta < 1.25$ for the case of adaptation to the Waymo dataset. It confirms that the performance crucially depends on the number of adapted parameters because of a conflict between the catastrophic forgetting and adapting to new domains. In practical usage scenarios, therefore, the degree or amount of adaptation on the MDE network should be judiciously determined. For the MDE network of MonoDepth2, adapting only the parameters in the BN layers of the encoder (as described previously in Section 4.1) turned out to excel in almost all performance metrics in our experimental setup. Extended results of this ablation study for other MDE metrics and other MDE networks are available in Appendix H.4.

## 5 CONCLUSIONS

In this work, we introduced PITTA, an innovative and novel TTA framework for MDE, markedly outperforming the prior SOTA methods in online adaptation of pretrained MDE network without accesses to source datasets and camera pose information. By leveraging instance-wise masks for dynamic objects in a novel and elegant manner, our method substantially enhanced the adaptability of MDE network during TTA. On top of this, we also devised effective edge extraction methodology and loss function design to further improve the MDE network's adaptability. With these strategies, our framework in turn exhibited superior performance and better effectiveness over other competing methods, as substantiated by thorough and extensive experimental validation on Driving Stereo and Waymo datasets, thereby underlining its good potential and wide applicability in TTA for MDE. Ablation studies further confirm efficacy and superiority of our model design choices. The scope and applicability of our TTA framework may be confined to MDE tasks, and thus, it is deserved to study the extendability and generality of our framework to other tasks as further works.

REPRODUCIBILITY STATEMENT

We have made extensive efforts to ensure the reproducibility of our work. The complete implementation of our framework, **PITTA**, is provided as source code in the supplementary material. Details of the model architectures (Section 3.1, Appendix D), hyper-parameter settings (Section 4.1, Appendix F), optimization details, and computational resources (Section 4.1) are explicitly described in the main paper and appendix. All evaluation protocols, datasets used (DrivingStereo, Waymo), and performance metrics (Section 4.1, Appendix E) are clearly documented. The core components of our proposed methodology, including the pose-agnostic paradigm, instance-aware masking, edge extraction, and loss function design, are described in Section 3, with the corresponding pseudo-code provided in Algorithm 1 for clarity. Extended experimental results for various MDE backbones and comprehensive ablation studies are reported in Appendix H. Collectively, these resources provide the necessary information to facilitate faithful reproduction and independent verification of our results.

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

# Supplementary Material

TABLE OF CONTENTS

## A    POOR ADAPTABILITY OF POSE ESTIMATION NETWORK WITH SfM ASSUMPTION

The naive approach of adapting a pose estimation network under the SfM assumption—such as in CoMoDA Kuznietsov et al. (2021)—is based on the following photometric loss function Godard et al. (2019); Guizilini et al. (2020); Zhang et al. (2023); Zhao et al. (2022):

$$L_{\mathrm{p}} = \sum_{(x,y) \in \mathcal{H} \times \mathcal{W}} \left\| I(x,y) - \hat{I}(x,y) \right\|_1 \tag{12}$$

where $\hat{I}$ is a reconstructed version of the input image $I$ using the pose estimation network. In such a naive approach, the quality of the reconstructed image $\hat{I}$ thus critically affects the adaptability of the pose estimation network (as well as the MDE network). Unfortunately, however, the naive approach could lead to improper adaptations during TTA, even exhibiting performance inferior to that of no adaptation, as demonstrated in Table 6 and Figure 3. Specifically, as pointed out in Li et al. (2023), such approach makes error accumulated, which eventually results in deterioration in both adaptability and performance of the pose estimation network on a long-term timescale.

Table 6: Results of CoMoDA, CoTTA, and ActMAD in adapting pose estimation networks pre-trained on KITTI dataset to DrivingStereo dataset for test case under sunny whether condition. Other conventions are the same as in Table 1 of the main text.

| Weather | Method | AbsRel ↓ | SqRel ↓ | RMSE ↓ | RMSElog ↓ | $\delta < 1.25$ ↑ | $\delta < 1.25^2$ ↑ | $\delta < 1.25^3$ ↑ |
|---|---|---|---|---|---|---|---|---|
| | No adaptation | 0.177 | 2.103 | 8.209 | 0.240 | 0.782 | 0.925 | 0.968 |
| | CoMoDA | 0.514 | 8.587 | 17.746 | 0.644 | 0.300 | 0.539 | 0.720 |
| Sunny | CoTTA | 0.184 | 2.090 | 8.305 | 0.248 | 0.767 | 0.920 | 0.964 |
| | ActMAD | 0.192 | 2.186 | 8.813 | 0.265 | 0.736 | 0.911 | 0.960 |

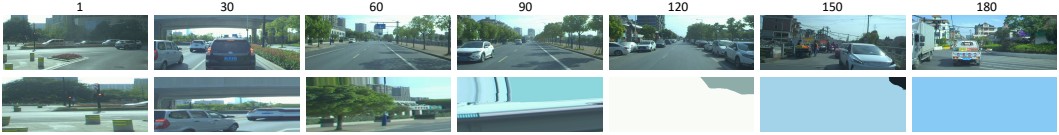

Figure 3: Poor adaptability of pose estimation network of CoMoDA with SfM assumption. Figures in upper and bottom rows represent original and reconstruction versions of input images, respectively. Numbers indicate time steps for adapting the pose estimation network.

## B    DETAILS OF RELATED WORKS

### B.1    DETAILS OF RELATED WORKS ON MDE

#### B.1.1    MULTI-SCALE MDE NETWORK

In Eigen et al. (2014), a multi-scale deep network for MDE was designed based on CNNs. This was the first deep learning approach to single image depth estimation based on supervised learning with scale-invariant loss. It thus requires ground-truth depth maps for training. The core architecture comprises coarse and fine networks. The coarse network predicts a global, coarse depth map, while the fine network refines the output of the coarse network locally to produce a finer and more detailed depth map. The method aimed at combining information from both global and local views depending on the importance on accurate depth estimation.

#### B.1.2    NEWCRFS

NewCRFs Yuan et al. (2022) is a supervised MDE model based on Transformer. This scheme was inspired by traditional methods that used Conditional Random Fields (CRFs) to model relationships between pixels to improve depth consistency and boundary sharpness. However, the traditional

methods are computationally expensive for dense prediction. The key method of NewCRFs is the introduction of Neural Window Fully-Connected CRFs (FC-CRFs) by making use of smaller, overlapping windows. NewCRFs used a Vision Transformer (ViT) as the encoder (for capturing global context) and the Neural Window FC-CRFs module as the decoder.

### B.1.3 ADABINS

AdaBins Bhat et al. (2021) introduced a novel approach to depth prediction based on adaptive binning. Instead of directly regressing depth values or classifying them into fixed depth ranges, it divided the depth interval into a number of bins whose centers are dynamically computed for each input image. The depth is finally predicted as a linear combination of the adaptive bin centers, which helps avoid the sharp discontinuities that often appear in classification-based methods. The network architecture was designed by incorporating a Transformer-based block (Mini-ViT) to perform a global statistical analysis of the features. This helps overcome the limited receptive field of traditional CNNs and refine the output at high resolution.

### B.1.4 SGDEPTH

SGDepth Guizilini et al. (2020) addressed the challenge of moving dynamic objects in self-supervised depth estimation. The framework adopted a multi-task learning approach that jointly optimizes semantic segmentation and depth estimation across different domains, using semantic information to guide the depth estimation process. SGDepth introduced a novel cross-domain training technique with task-specific network heads, allowing the supervised semantic segmentation branch to be trained on one dataset while the self-supervised depth estimation is trained on another. It employed gradient scaling to enable effective cross-domain training, ensuring optimally learned task-specific weights in the respective decoders. A key innovation of SGDepth is its semantic masking scheme that identifies dynamic class objects through segmentation and selectively excludes moving objects from contaminating the photometric loss, while a detection method for non-moving dynamic objects allows the network to still learn depth for static instances of typically dynamic objects like parked cars. This mutually beneficial approach results in improved accuracy for both depth estimation and semantic segmentation tasks.

### B.1.5 HR-DEPTH

HRDepth Lyu et al. (2021) focused on improving depth estimation accuracy by specifically addressing inaccuracies in large gradient regions. The authors discovered that as image resolution increases, bilinear interpolation errors gradually disappear, but depth estimation in boundary regions remained challenging. To solve this problem, HR-Depth incorporated two key innovations: a redesigned skip-connection architecture that better preserves and utilizes high-resolution features with rich semantic information, and a feature fusion Squeeze-and-Excitation (fSE) module that efficiently fuses features while reducing the overall parameter count. The framework used a U-Net based architecture with dense skip connections that aggregate features across multiple scales, helping reduce the semantic gap between encoder and decoder features.

### B.1.6 MONOVIT

MonoViT Zhao et al. (2022) addressed the issue in the limited receptive field of traditional CNNs by integrating Transformer blocks to model long-range relationships between pixels, enabling a global receptive field that captures detailed scene structures. The architecture comprises a depth encoder that uses both convolutions for modeling local features and parallel Transformer blocks to extract global contexts, with a feature fusion mechanism that effectively combines both types of information. MonoViT employed a multi-path approach with multiple parallel blocks, including a Multi-Scale Patch Embedding (MSPE) layer that extracts various-sized visual tokens and a Global-to-Local Feature Interaction (GLFI) module to enhance feature modeling.

### B.1.7 LITE-MONO

Lite-Mono Zhang et al. (2023) is a lightweight architecture designed for self-supervised MDE that achieves impressive accuracy while using significantly fewer parameters than other comparable

models. It employed a hybrid architecture that efficiently combines CNNs and Transformers through two main components: the Consecutive Dilated Convolutions (CDC) module and the Local-Global Features Interaction (LGFI) module. The CDC module extracts rich multi-scale local features using dilated convolutions with different dilation rates to expand the receptive field without adding parameters, while the LGFI module incorporates Transformer-based self-attention to encode long-range global information using cross-covariance attention in the channel dimension to reduce computational complexity. The encoder consists of four stages with CDC blocks and LGFI modules, while the decoder increases spatial dimensions using bilinear upsampling and skip connections from the encoder.

### B.1.8 DEPTHANYTHING V2

DepthAnything V2 Yang et al. (2024) that leverages a teacher-student paradigm, where its highly capable teacher model facilitates the training of robust student models suitable for downstream tasks. It is built on discriminative models while addressing weaknesses observed in generative models. V2 models are significantly more efficient and have fewer parameters compared to generative model based methods. V2 solely trains its student models on large-scale pseudo-labeled real images generated by its high-quality synthetic-trained teacher model. This process is key to transferring the precision learned from synthetic data to diverse real-world distributions and improving generalization, especially for smaller student models. By scaling up the teacher model capacity and leveraging high-quality synthetic training data, the framework effectively addresses the fundamental domain gap between controlled synthetic environments and complex real-world imaging conditions.

### B.2 DETAILS OF RELATED WORKS ON TTA FOR MDE

### B.2.1 CoMoDA

CoMoDA Kuznietsov et al. (2021) has been proposed for continuous adaptation of a pretrained MDE model on a test video in real-time. A key approach is the replay buffer mechanism, which stores samples from the source dataset. It serves as a critical component that prevents the model from overfitting to short video. By augmenting each adaptation iteration with both current video frames and randomly sampled frames from the replay buffer, the method effectively stabilizes the adaptation process, allowing for consistent performance improvements throughout longer sequences without degradation.

### B.2.2 ADA-DEPTH

Ada-Depth Li et al. (2023) is a source-free, online test-time domain adaptation method for MDE in continuously changing environments. It addressed challenges in absolute scale inference, inaccurate pseudo-labels, and catastrophic forgetting. The framework introduced an alignment scheme that geometrically aligns input features. This process leverages intrinsic camera parameters from both domains, including focal length ratios and camera heights, to correct the absolute scale inference on target data. Furthermore, the framework implemented a pseudo label consistency checking mechanism to identify and select confident pixels, thereby enhancing pseudo label quality during adaptation. Combined with regularization and self-training schemes, it enables stable long-term adaptation that maintains scale-awareness while significantly improving inference quality on target domains.

## C DETAILED COMPUTATION PROCEDURE OF OUR FRAMEWORK

In Algorithm 1, we present the detailed computation procedure of our TTA framework, PITTA.

## D DETAILS OF PRIMARY NETWORKS ADOPTED IN OUR FRAMEWORK

### D.1 DETAILS OF MONODEPTH2

MonoDepth2 Godard et al. (2019) is a self-supervised learning approach for training depth estimation models from monocular videos. It leverages the geometric relationship between consecutive

---

**Algorithm 1** Proposed PITTA Framework

---

**Input:** Sequences of single RGB images from a monocular camera.
**Output:** Adapted parameter set $\theta$ for a pretrained MDE network.
1: **loop**
2:  Predict a depth map $D$ for a single monocular image $I$ using the MDE network.
3:  Predict a segmentation mask $S$ for the input image $I$ from a panoptic segmentation network.
4:  Perform instance-aware masking on $D$ to obtain masked depth maps $\{\tilde{D}_j\}$ by equation 4.
5:  Compute edge maps $\partial I$ and $\partial D$ by equation 19 and equation 6, respectively.
6:  Perform median filtering on $\{\tilde{D}_j\}$ to create STATE labels $\{\tilde{D}'_j\}$ by equation 7.
7:  Calculate depth-refining loss $L_\mathrm{d}$ and edge-guided loss $L_\mathrm{e}$ by equation 8 and equation 9, respectively.
8:  Compute the overall loss $L$ by equation 10 and adapt $\theta$ according to equation 11.
9: **end loop**

---

frames. By estimating the ego-motion (camera's movement) between frames and the depth of the scene, it tries to reconstruct an image in a specific frame from images in another frames. The difference between the reconstructed image and the actual image (the photometric reprojection error) serves as the primary training signal or pseudo-label. The networks are learned to minimize this error, implicitly learning about depth and motion. In MonoDepth2, there are two types of networks: MDE network and pose estimation network. In our proposed method, the MDE network of MonoDepth2 is adopted, which takes a single RGB image from a monocular camera as input and predicts a single-channel depth map, where each pixel's value represents its estimated distance from the camera. The MDE network of MonoDepth2 also employs an encoder-decoder architecture, often with a ResNet backbone for feature extraction in the encoder and a U-Net style decoder to produce the depth map at the original resolution.

### D.2 DETAILS OF MASK2FORMER

Mask2Former Cheng et al. (2022) is a novel transformer-based architecture for universal image segmentation, capable of dealing with panoptic, instance, and semantic segmentation tasks with a single model. Its core technology is the masked attention mechanism. Instead of performing global self-attention, Mask2Former constrains the cross-attention within the regions of predicted masks. This allows the network to extract more localized and relevant features, leading to significant performance improvements and improved efficiency compared to previous approaches like MaskFormer. By unifying these three fundamental segmentation tasks, Mask2Former achieved state-of-the-art results on multiple challenging datasets, demonstrating its robustness. Mask2Former includes the following main components:

- Backbone feature extractor: This extracts low-resolution feature maps from the input image, which can be a CNN like ResNet or a Transformer-based network like Swin Transformer.

- Pixel decoder: This upsamples the low-resolution features from the backbone freature extractor to generate high-resolution per-pixel embeddings, which helps in producing detailed segmentation masks.

- Masked attention: Instead of the standard cross-attention mechanism that attends to all pixels in the image, Mask2Former employed the masked attention. This constrains the cross-attention within the predicted mask regions, allowing the decoder to efficiently extract localized features relevant to each object or semantic category and leading to faster convergence with improved performance.

## E DETAILS OF MDE PERFORMANCE METRICS

In the experimentation, the quality of MDE is measured using the following quantitative metrics:

- Threshold accuracy: This is the percentage of pixels whose maximum ratio between predicted depth value and ground-truth depth value (i.e, $\delta \triangleq \max\left(\frac{D_\mathrm{gt}(x,y)}{D(x,y)}, \frac{D(x,y)}{D_\mathrm{gt}(x,y)}\right)$) is

below a threshold $\tau$, i.e.,

$$\text{Threshold accuracy} = \left| \left\{ (x,y) \in \mathcal{T} : \max \left( \frac{D_{\text{gt}}(x,y)}{D(x,y)}, \frac{D(x,y)}{D_{\text{gt}}(x,y)} \right) < \tau \right\} \right| \quad (13)$$

where $D(x,y)$ and $D_{\text{gt}}(x,y)$ denote the predicted and ground-truth depth values for pixel $(x,y)$, respectively. Also, $\mathcal{T}$ denotes a collection of pixels for which the ground-truth depth values are available, with cardinality $T$. Typically, the threshold value is chosen as $\tau \in \{1.25, 1.25^2, 1.25^3\}$.

- AbsRel: This is defined as the absolute difference between predicted and ground-truth depth values relative to the ground-truth value, i.e.,

$$\text{AbsRel} = \frac{1}{T} \sum_{(x,y) \in \mathcal{T}} \frac{\left| D_{\text{gt}}(x,y) - D(x,y) \right|}{D_{\text{gt}}(x,y)}. \quad (14)$$

- SqRel: This is defined as the squared difference between predicted and ground-truth depth values relative to the ground-truth value, i.e.,

$$\text{SqRel} = \frac{1}{T} \sum_{(x,y) \in \mathcal{T}} \frac{\left( D_{\text{gt}}(x,y) - D(x,y) \right)^2}{D_{\text{gt}}(x,y)}. \quad (15)$$

- RMSE: This is defined as the square root of the average squared difference between predicted and ground-truth depth values, i.e.,

$$\text{RMSE} = \sqrt{\frac{1}{T} \sum_{(x,y) \in \mathcal{T}} \left( D_{\text{gt}}(x,y) - D(x,y) \right)^2}. \quad (16)$$

- RMSElog: This corresponds to the RMSE between predicted and ground-truth depth values on a log scale, i.e.,

$$\text{RMSElog} = \sqrt{\frac{1}{T} \sum_{(x,y) \in \mathcal{T}} \left( \log D_{\text{gt}}(x,y) - \log D(x,y) \right)^2}. \quad (17)$$

## F  DETAILED EXPERIMENT SETTING

In the experiments, we adapt the MDE network according to Algorithm 1 in the main text with $\alpha = 0.0001$, $\lambda = 0.2$, and $s = 5$. Also, the weights for extracting the edges maps $\partial I$ and $\partial D$ are, respectively, chosen as

$$U(x,y) = \frac{A(x,y)}{\max_{(p,q) \in \mathcal{H} \times \mathcal{W}} A(p,q)}, \qquad V(x,y) = 1 \quad (18)$$

for $\forall (x,y) \in \mathcal{H} \times \mathcal{W}$, where

$$A(x,y) = \big| \bar{I}(\min\{x+1, H-1\}, y) + \bar{I}(\max\{x-1, 0\}, y) + \bar{I}(x, \min\{y+1, W-1\})$$
$$+ \bar{I}(x, \max\{y-1, 0\}) - 4\bar{I}(x,y) \big|. \quad (19)$$

The Adam optimizer is used to perform the stochastic gradient descent (SGD) update in Eq. (11) of the main text.

## G  DATASET PREPROCESSING

DrivingStereo dataset contains monocular images with height of 800 and width of 1762 (i.e., $800 \times 1762$) on each channel, and Waymo dataset contains monocular images with height of 1280 and width of 1920 (i.e., $1280 \times 1920$) on each channel. Prior to the main computation procedures in Algorithm 1 of the main text, the images in the DrivingStereo or Waymo dataset are preprocessed: they are resized by bilinear interpolation and their pixel values are normalized via maxmin scaling. In Table 7, detailed specifications on resized images are listed for various MDE networks. The number of channels are kept to be the same during this preprocessing.

Table 7: Specification of image resizing in dataset preprocessing.

| MDE Net | Size of resized image |
|---|---|
| MonoDepth2 | $512 \times 192$ |
| SGDepth | $512 \times 192$ |
| HR-Depth | $1280 \times 384$ |
| MonoViT | $512 \times 192$ |
| Lite-Mono | $512 \times 192$ |
| DepthAnything V2 | $518 \times 196$ |
| AdaBins | $1216 \times 352$ |
| NewCRFs | $1236 \times 376$ |

## H SUPPLEMENTARY RESULTS

### H.1 EXPERIMENTAL RESULTS FOR DIVERSE MDE NETWORKS

Tables 9–15 show the full results of our framework for adaptations of MDE networks in various self-supervised approaches—SGDepth Guizilini et al. (2020), HR-Depth Lyu et al. (2021), Lite-Mono Zhang et al. (2023), and MonoViT Zhao et al. (2022), respectively. Also, Tables 13 and 14 show the results of our framework for adaptations of MDE networks in two supervised approaches—AdaBins Bhat et al. (2021) and NewCRFs Yuan et al. (2022), respectively. The results in Tables 9–14 completely prove the versatility, universality, and efficacy of our TTA strategy for various types of MDE networks.

### H.2 EXPERIMENTAL RESULTS FOR MASK2FORMER WITH DIFFERENT BACKBONES

In Table 16, we additionally demonstrate the performance of our framework over various network configurations of Mask2Former with different backbones based on Swin Transformer—Swin-Tiny, Swin-Small, and Swin-Base. It further confirms the robustness and stability of our framework to the performance or variation of the panoptic segmentation method.

### H.3 EXTENDED RESULTS OF ABLATION STUDY ON TRADEOFF PARAMETER $\lambda$

The full results of our ablation study on $\lambda$ for all MDE metrics are presented in Table 17. It can be seen that our framework performs best with $\lambda = 0.2$ in most test cases. This is the rationale behind selecting $\lambda = 0.2$ in the experimentation.

### H.4 EXTENDED RESULTS OF ABLATION STUDY ON THE NUMBER OF ADAPTED PARAMETERS

The full results of the ablation study on the number of adapted parameters in the encoder of other MDE networks for all MDE metrics are shown throughout Tables 18–22. The results in 18–22 consistently show that our framework demonstrates best adaptability across various MDE networks when adapting only the normalization layers (i.e., BN 100% or Norm 100%). Adapting other parts of networks can lead to significant performance degradation on account primarily of severe catastrophic forgetting and/or insufficient adaptability to new domains.

Table 8: Results of our proposed method and other competing methods applied to adaptation of MDE network of MonoDepth2 pretrained on KITTI dataset to DrivingStereo dataset. In each test case, numbers in **bold** and underline indicate the best performance and the second best performance, respectively. Also, "All" indicates a test case concatenating cloudy, foggy, rainy, and sunny weather conditions in turn.

| Weather | Method | AbsRel ↓ | SqRel ↓ | RMSE ↓ | RMSElog ↓ | $\delta < 1.25$ ↑ | $\delta < 1.25^2$ ↑ | $\delta < 1.25^3$ ↑ |
|---|---|---|---|---|---|---|---|---|
|  | No adaptation | 0.143 | 1.952 | 9.817 | 0.218 | 0.812 | 0.937 | 0.974 |
|  | CoMoDA | 0.570 | 11.179 | 21.563 | 0.665 | 0.245 | 0.463 | 0.655 |
|  | CoTTA | 0.175 | 2.344 | 9.899 | 0.243 | 0.750 | 0.920 | 0.969 |
| Foggy | ActMAD | 0.176 | 2.403 | 11.157 | 0.262 | 0.730 | 0.902 | 0.957 |
|  | Ada-Depth | 0.317 | 5.048 | 10.642 | 0.387 | 0.562 | 0.791 | 0.896 |
|  | DepthAnythingV2 | 0.164 | 2.741 | 12.891 | 0.278 | 0.736 | 0.880 | 0.944 |
|  | **PITTA (ours)** | **0.127** | **1.579** | **8.600** | **0.195** | **0.840** | **0.951** | **0.980** |
|  | No adaptation | 0.170 | 2.211 | 8.453 | **0.232** | **0.781** | **0.931** | **0.973** |
|  | CoMoDA | 0.475 | 7.903 | 18.030 | 0.619 | 0.308 | 0.553 | 0.734 |
|  | CoTTA | 0.189 | 2.413 | 8.597 | 0.247 | 0.744 | 0.914 | 0.967 |
| Cloudy | ActMAD | 0.170 | 1.956 | 8.784 | 0.247 | 0.764 | 0.923 | 0.968 |
|  | Ada-Depth | 0.236 | 3.566 | 10.611 | 0.302 | 0.648 | 0.866 | 0.945 |
|  | DepthAnythingV2 | 0.183 | 2.356 | 10.962 | 0.283 | 0.720 | 0.895 | 0.952 |
|  | **PITTA (ours)** | **0.166** | **1.866** | **8.148** | **0.232** | **0.781** | 0.929 | 0.972 |
|  | No adaptation | 0.245 | 3.641 | 12.282 | 0.310 | 0.600 | 0.852 | 0.945 |
|  | CoMoDA | 0.701 | 18.848 | 22.759 | 0.727 | 0.303 | 0.497 | 0.653 |
|  | CoTTA | 0.225 | 3.127 | 11.057 | 0.286 | 0.638 | 0.875 | 0.960 |
| Rainy | ActMAD | 0.256 | 3.631 | 12.399 | 0.328 | 0.590 | 0.818 | 0.936 |
|  | Ada-Depth | 0.322 | 5.623 | 10.782 | 0.387 | 0.572 | 0.794 | 0.897 |
|  | DepthAnythingV2 | 0.208 | 3.105 | 12.480 | 0.293 | 0.657 | 0.860 | 0.946 |
|  | **PITTA (ours)** | **0.195** | **2.532** | **10.315** | **0.254** | **0.685** | **0.905** | **0.973** |
|  | No adaptation | **0.177** | 2.103 | 8.209 | **0.240** | **0.782** | **0.925** | **0.968** |
|  | CoMoDA | 0.515 | 8.306 | 18.065 | 0.643 | 0.289 | 0.522 | 0.711 |
|  | CoTTA | 0.187 | 2.135 | 8.341 | 0.250 | 0.761 | 0.919 | 0.963 |
| Sunny | ActMAD | 0.192 | 2.186 | 8.813 | 0.265 | 0.736 | 0.911 | 0.960 |
|  | Ada-Depth | 0.212 | 3.431 | 10.011 | 0.279 | 0.715 | 0.883 | 0.948 |
|  | DepthAnythingV2 | 0.194 | 2.473 | 11.197 | 0.300 | 0.707 | 0.884 | 0.944 |
|  | **PITTA (ours)** | **0.177** | **1.938** | **8.138** | 0.244 | **0.782** | 0.923 | 0.963 |
|  | No adaptation | 0.181 | 2.446 | 9.536 | 0.247 | 0.749 | 0.913 | 0.965 |
|  | CoMoDA | 0.577 | 11.736 | 20.289 | 0.673 | 0.277 | 0.497 | 0.678 |
|  | CoTTA | 0.200 | 2.619 | 9.657 | 0.265 | 0.710 | 0.899 | 0.961 |
| All | ActMAD | 0.172 | 2.190 | 9.543 | 0.244 | **0.758** | 0.913 | 0.967 |
|  | Ada-Depth | 0.272 | 4.417 | 10.512 | 0.339 | 0.624 | 0.833 | 0.921 |
|  | DetphAnythingV2 | 0.187 | 2.669 | 11.883 | 0.289 | 0.705 | 0.880 | 0.946 |
|  | **PITTA (ours)** | **0.171** | **2.016** | **9.200** | **0.241** | **0.758** | **0.918** | **0.968** |

Table 9: Results of our proposed method applied to adaptation of MDE network of SGDepth pretrained on KITTI dataset to DrivingStereo dataset. Other conventions are the same as in Table 1 of the main text.

| Weather | Method | AbsRel ↓ | SqRel ↓ | RMSE ↓ | RMSElog ↓ | $\delta < 1.25$ ↑ | $\delta < 1.25^2$ ↑ | $\delta < 1.25^3$ ↑ |
|---|---|---|---|---|---|---|---|---|
| Foggy | No adaptation | 0.233 | 3.453 | 13.553 | 0.338 | 0.603 | 0.841 | 0.924 |
|  | **PITTA (ours)** | 0.127 | 1.621 | 9.023 | 0.204 | 0.830 | 0.943 | 0.975 |
| Cloudy | No adaptation | 0.185 | 2.302 | 9.949 | 0.279 | 0.727 | 0.902 | 0.955 |
|  | **PITTA (ours)** | 0.161 | 1.830 | 8.400 | 0.234 | 0.785 | 0.928 | 0.971 |
| Rainy | No adaptation | 0.320 | 5.249 | 14.195 | 0.389 | 0.528 | 0.772 | 0.891 |
|  | **PITTA (ours)** | 0.195 | 2.624 | 10.646 | 0.257 | 0.692 | 0.902 | 0.970 |
| Sunny | No adaptation | 0.207 | 2.527 | 9.738 | 0.290 | 0.725 | 0.893 | 0.948 |
|  | **PITTA (ours)** | 0.172 | 1.864 | 8.229 | 0.243 | 0.785 | 0.924 | 0.965 |
| All | **PITTA (ours)** | 0.177 | 2.245 | 9.908 | 0.257 | 0.745 | 0.904 | 0.961 |

Table 10: Results of our proposed method applied to adaptation of MDE network of HR-Depth pretrained on KITTI dataset to DrivingStereo dataset. Other conventions are the same as in Table 1 of the main text.

| Weather | Method | AbsRel ↓ | SqRel ↓ | RMSE ↓ | RMSElog ↓ | $\delta < 1.25$ ↑ | $\delta < 1.25^2$ ↑ | $\delta < 1.25^3$ ↑ |
|---|---|---|---|---|---|---|---|---|
| Foggy | No adaptation | 0.139 | 1.822 | 9.472 | 0.207 | 0.813 | 0.944 | 0.981 |
| | **PITTA (ours)** | 0.129 | 1.500 | 8.656 | 0.194 | 0.828 | 0.951 | 0.983 |
| Cloudy | No adaptation | 0.161 | 2.112 | 8.172 | 0.224 | 0.801 | 0.935 | 0.976 |
| | **PITTA (ours)** | 0.169 | 1.833 | 8.239 | 0.237 | 0.772 | 0.927 | 0.968 |
| Rainy | No adaptation | 0.274 | 4.554 | 12.833 | 0.335 | 0.588 | 0.828 | 0.924 |
| | **PITTA (ours)** | 0.193 | 2.514 | 10.732 | 0.256 | 0.687 | 0.907 | 0.973 |
| Sunny | No adaptation | 0.164 | 1.694 | 7.617 | 0.225 | 0.783 | 0.938 | 0.977 |
| | **PITTA (ours)** | 0.171 | 1.711 | 7.970 | 0.240 | 0.787 | 0.924 | 0.964 |
| All | **PITTA (ours)** | 0.169 | 1.865 | 9.022 | 0.255 | 0.776 | 0.913 | 0.959 |

Table 11: Results of our proposed method applied to adaptation of MDE network of Lite-Mono pretrained on KITTI dataset to DrivingStereo dataset. Other conventions are the same as in Table 1 of the main text.

| Weather | Method | AbsRel ↓ | SqRel ↓ | RMSE ↓ | RMSElog ↓ | $\delta < 1.25$ ↑ | $\delta < 1.25^2$ ↑ | $\delta < 1.25^3$ ↑ |
|---|---|---|---|---|---|---|---|---|
| Foggy | No adaptation | 0.139 | 1.780 | 9.369 | 0.209 | 0.810 | 0.944 | 0.980 |
| | **PITTA (ours)** | 0.125 | 1.509 | 8.556 | 0.194 | 0.837 | 0.951 | 0.981 |
| Cloudy | No adaptation | 0.164 | 2.024 | 8.473 | 0.234 | 0.780 | 0.932 | 0.973 |
| | **PITTA (ours)** | 0.157 | 1.744 | 8.293 | 0.229 | 0.788 | 0.932 | 0.972 |
| Rainy | No adaptation | 0.215 | 3.294 | 11.980 | 0.284 | 0.655 | 0.876 | 0.958 |
| | **PITTA (ours)** | 0.211 | 2.917 | 11.085 | 0.271 | 0.662 | 0.888 | 0.965 |
| Sunny | No adaptation | 0.184 | 2.166 | 8.384 | 0.248 | 0.766 | 0.918 | 0.965 |
| | **PITTA (ours)** | 0.171 | 1.844 | 7.970 | 0.240 | 0.787 | 0.924 | 0.964 |
| All | **PITTA (ours)** | 0.173 | 2.079 | 9.406 | 0.243 | 0.750 | 0.915 | 0.968 |

Table 12: Results of our proposed method applied to adaptation of MDE network of MonoViT pretrained on KITTI dataset to DrivingStereo dataset. Other conventions are the same as in Table 1 of the main text.

| Weather | Method | AbsRel ↓ | SqRel ↓ | RMSE ↓ | RMSElog ↓ | $\delta < 1.25$ ↑ | $\delta < 1.25^2$ ↑ | $\delta < 1.25^3$ ↑ |
|---|---|---|---|---|---|---|---|---|
| Foggy | No adaptation | 0.109 | 1.205 | 7.760 | 0.167 | 0.870 | 0.967 | 0.990 |
| | **PITTA (ours)** | 0.117 | 1.368 | 8.699 | 0.186 | 0.844 | 0.951 | 0.984 |
| Cloudy | No adaptation | 0.141 | 1.621 | 7.546 | 0.201 | 0.831 | 0.948 | 0.981 |
| | **PITTA (ours)** | 0.134 | 1.413 | 7.542 | 1.999 | 0.831 | 0.950 | 0.982 |
| Rainy | No adaptation | 0.179 | 2.183 | 9.641 | 0.236 | 0.724 | 0.924 | 0.979 |
| | **PITTA (ours)** | 0.175 | 2.078 | 9.769 | 0.236 | 0.725 | 0.924 | 0.979 |
| Sunny | No adaptation | 0.150 | 1.609 | 7.648 | 0.211 | 0.815 | 0.943 | 0.979 |
| | **PITTA (ours)** | 0.141 | 1.425 | 7.528 | 0.207 | 0.826 | 0.948 | 0.979 |
| All | **PITTA (ours)** | 0.150 | 1.550 | 7.609 | 0.213 | 0.814 | 0.938 | 0.977 |

Table 13: Results of our proposed method applied to adaptation of MDE network of AdaBins pretrained on KITTI dataset to DrivingStereo dataset. Other conventions are the same as in Table 1 of the main text.

| Weather | Method | AbsRel ↓ | SqRel ↓ | RMSE ↓ | RMSElog ↓ | $\delta < 1.25$ ↑ | $\delta < 1.25^2$ ↑ | $\delta < 1.25^3$ ↑ |
|---|---|---|---|---|---|---|---|---|
| Foggy | No adaptation | 0.474 | 8.094 | 18.908 | 0.559 | 0.276 | 0.539 | 0.757 |
| | **PITTA (ours)** | 0.120 | 1.537 | 9.135 | 0.201 | 0.840 | 0.945 | 0.979 |
| Cloudy | No adaptation | 0.328 | 4.680 | 14.638 | 0.455 | 0.451 | 0.737 | 0.871 |
| | **PITTA (ours)** | 0.164 | 1.610 | 7.841 | 0.231 | 0.786 | 0.929 | 0.971 |
| Rainy | No adaptation | 0.631 | 14.584 | 20.607 | 0.666 | 0.312 | 0.526 | 0.685 |
| | **PITTA (ours)** | 0.200 | 2.633 | 11.607 | 0.266 | 0.680 | 0.896 | 0.969 |
| Sunny | No adaptation | 0.297 | 4.418 | 13.905 | 0.423 | 0.531 | 0.776 | 0.883 |
| | **PITTA (ours)** | 0.169 | 1.657 | 7.789 | 0.235 | 0.793 | 0.925 | 0.967 |
| All | **PITTA (ours)** | 0.190 | 2.123 | 8.940 | 0.261 | 0.744 | 0.915 | 0.963 |

Table 14: Results of our proposed method applied to adaptation of MDE network of NewCRFs pretrained on KITTI dataset to DrivingStereo dataset. Other conventions are the same as in Table 1 of the main text.

| Weather | Method | AbsRel ↓ | SqRel ↓ | RMSE ↓ | RMSElog ↓ | $\delta < 1.25$ ↑ | $\delta < 1.25^2$ ↑ | $\delta < 1.25^3$ ↑ |
|---|---|---|---|---|---|---|---|---|
| Foggy | No adaptation | 0.139 | 1.641 | 9.081 | 0.218 | 0.806 | 0.936 | 0.971 |
| | **PITTA (ours)** | 0.137 | 1.612 | 8.990 | 0.215 | 0.810 | 0.938 | 0.972 |
| Cloudy | No adaptation | 0.139 | 1.429 | 7.208 | 0.201 | 0.827 | 0.951 | 0.980 |
| | **PITTA (ours)** | 0.139 | 1.429 | 7.208 | 0.201 | 0.827 | 0.951 | 0.980 |
| Rainy | No adaptation | 0.239 | 3.025 | 10.740 | 0.295 | 0.631 | 0.856 | 0.951 |
| | **PITTA (ours)** | 0.234 | 2.934 | 10.538 | 0.290 | 0.638 | 0.864 | 0.954 |
| Sunny | No adaptation | 0.141 | 1.210 | 6.921 | 0.202 | 0.823 | 0.948 | 0.980 |
| | **PITTA (ours)** | 0.142 | 1.215 | 6.904 | 0.202 | 0.822 | 0.949 | 0.981 |
| All | **PITTA (ours)** | 0.155 | 1.302 | 6.904 | 0.208 | 0.798 | 0.948 | 0.982 |

Table 15: Results of our proposed method applied to adaptation of MDE network of DepthAnythinV2 pretrained on vKITTI dataset to DrivingStereo dataset. Other conventions are the same as in Table 1 of the main text.

| Weather | Method | AbsRel ↓ | SqRel ↓ | RMSE ↓ | RMSElog ↓ | $\delta < 1.25$ ↑ | $\delta < 1.25^2$ ↑ | $\delta < 1.25^3$ ↑ |
|---|---|---|---|---|---|---|---|---|
| Foggy | No adaptation | 0.164 | 2.741 | 12.891 | 0.278 | 0.736 | 0.880 | 0.944 |
| | **PITTA (ours)** | 0.163 | 2.750 | 12.790 | 0.277 | 0.744 | 0.880 | 0.944 |
| Cloudy | No adaptation | 0.183 | 2.356 | 10.962 | 0.283 | 0.720 | 0.895 | 0.952 |
| | **PITTA (ours)** | 0.215 | 2.453 | 10.319 | 0.287 | 0.684 | 0.891 | 0.953 |
| Rainy | No adaptation | 0.208 | 3.105 | 12.480 | 0.293 | 0.657 | 0.860 | 0.946 |
| | **PITTA (ours)** | 0.209 | 3.156 | 12.619 | 0.296 | 0.654 | 0.857 | 0.944 |
| Sunny | No adaptation | 0.194 | 2.473 | 11.197 | 0.300 | 0.707 | 0.884 | 0.944 |
| | **PITTA (ours)** | 0.194 | 2.498 | 11.322 | 0.302 | 0.708 | 0.882 | 0.943 |
| All | **PITTA (ours)** | 0.198 | 3.001 | 12.593 | 0.305 | 0.684 | 0.857 | 0.935 |

Table 16: Results of our proposed method for various network configurations of Mask2Former with different backbones based on Swin Transformer. Other conventions are the same as in Table 1 of the main text.

| Backbone | Weather | AbsRel ↓ | SqRel ↓ | RMSE ↓ | RMSElog ↓ | $\delta < 1.25 \uparrow$ | $\delta < 1.25^2 \uparrow$ | $\delta < 1.25^3 \uparrow$ |
|---|---|---|---|---|---|---|---|---|
| | Foggy | 0.127 | 1.578 | 8.598 | 0.195 | 0.840 | 0.951 | 0.980 |
| | Cloudy | 0.166 | 1.865 | 8.149 | 0.232 | 0.781 | 0.929 | 0.972 |
| Swin-Tiny | Rainy | 0.195 | 2.533 | 10.317 | 0.254 | 0.685 | 0.905 | 0.973 |
| | Sunny | 0.177 | 1.937 | 8.132 | 0.244 | 0.782 | 0.923 | 0.963 |
| | Waymo DA | 0.199 | 2.097 | 8.206 | 0.266 | 0.670 | 0.896 | 0.967 |
| | Foggy | 0.127 | 1.578 | 8.622 | 0.196 | 0.840 | 0.951 | 0.980 |
| | Cloudy | 0.168 | 1.912 | 8.169 | 0.233 | 0.779 | 0.928 | 0.972 |
| Swin-Small | Rainy | 0.196 | 2.540 | 10.031 | 0.255 | 0.683 | 0.904 | 0.974 |
| | Sunny | 0.178 | 1.995 | 8.173 | 0.245 | 0.779 | 0.922 | 0.963 |
| | Waymo DA | 0.201 | 2.162 | 8.214 | 0.266 | 0.671 | 0.896 | 0.967 |
| | Foggy | 0.127 | 1.578 | 8.622 | 0.196 | 0.840 | 0.951 | 0.980 |
| | Cloudy | 0.168 | 1.912 | 8.169 | 0.233 | 0.779 | 0.928 | 0.972 |
| Swin-Base | Rainy | 0.195 | 2.514 | 10.287 | 0.254 | 0.685 | 0.906 | 0.974 |
| | Sunny | 0.176 | 1.947 | 8.145 | 0.244 | 0.782 | 0.923 | 0.963 |
| | Waymo DA | 0.200 | 2.138 | 8.724 | 0.266 | 0.672 | 0.894 | 0.966 |

Table 17: Ablation study on $\lambda$ regarding tradeoff between two loss functions $L_\mathrm{d}$ and $L_\mathrm{e}$ for adaptations to DrivingStereo and Waymo datasets. "Foggy", "Cloudy", "Rainy", "Sunny" indicate test cases for the DrivingStereo dataset under foggy, cloudy, rainy, and sunny weather conditions, respectively. Other conventions are the same as in Table 1 of the main text.

| $\lambda$ | Dataset | AbsRel ↓ | SqRel ↓ | RMSE ↓ | RMSElog ↓ | $\delta < 1.25$ ↑ | $\delta < 1.25^2$ ↑ | $\delta < 1.25^3$ ↑ |
|---|---|---|---|---|---|---|---|---|
| | Foggy | 0.133 | 1.745 | 8.828 | 0.199 | 0.837 | 0.950 | 0.981 |
| | Cloudy | 0.175 | 2.108 | 8.315 | 0.273 | 0.773 | 0.925 | 0.971 |
| 0 | Rainy | 0.210 | 2.906 | 10.791 | 0.270 | 0.667 | 0.889 | 0.966 |
| | Sunny | 0.184 | 2.136 | 8.304 | 0.248 | 0.771 | 0.919 | 0.963 |
| | Waymo | 0.236 | 3.094 | 9.149 | 0.293 | 0.618 | 0.876 | 0.958 |
| | Foggy | 0.129 | 1.622 | 8.640 | 0.196 | 0.840 | 0.952 | 0.981 |
| | Cloudy | 0.169 | 1.930 | 8.164 | 0.232 | 0.782 | 0.928 | 0.972 |
| 0.1 | Rainy | 0.200 | 2.638 | 10.438 | 0.258 | 0.679 | 0.899 | 0.972 |
| | Sunny | 0.179 | 1.983 | 8.163 | 0.245 | 0.779 | 0.922 | 0.963 |
| | Waymo | 0.203 | 1.736 | 7.359 | 0.263 | 0.644 | 0.902 | 0.974 |
| | Foggy | 0.127 | 1.578 | 8.598 | 0.195 | 0.840 | 0.951 | 0.980 |
| | Cloudy | 0.166 | 1.865 | 8.149 | 0.232 | 0.781 | 0.929 | 0.972 |
| 0.2 | Rainy | 0.195 | 2.533 | 10.317 | 0.254 | 0.685 | 0.905 | 0.973 |
| | Sunny | 0.177 | 1.937 | 8.132 | 0.244 | 0.782 | 0.923 | 0.963 |
| | Waymo | 0.199 | 2.097 | 8.206 | 0.266 | 0.670 | 0.896 | 0.967 |
| | Foggy | 0.127 | 1.562 | 8.604 | 0.196 | 0.840 | 0.951 | 0.980 |
| | Cloudy | 0.165 | 1.846 | 8.164 | 0.232 | 0.782 | 0.929 | 0.971 |
| 0.3 | Rainy | 0.194 | 2.493 | 10.276 | 0.253 | 0.688 | 0.907 | 0.974 |
| | Sunny | 0.176 | 1.915 | 8.129 | 0.243 | 0.783 | 0.923 | 0.963 |
| | Waymo | 0.200 | 2.085 | 8.271 | 0.268 | 0.668 | 0.894 | 0.967 |
| | Foggy | 0.126 | 1.558 | 8.629 | 0.196 | 0.840 | 0.950 | 0.979 |
| | Cloudy | 0.165 | 1.835 | 8.176 | 0.232 | 0.782 | 0.929 | 0.971 |
| 0.4 | Rainy | 0.192 | 2.463 | 10.249 | 0.252 | 0.690 | 0.908 | 0.974 |
| | Sunny | 0.176 | 1.915 | 8.129 | 0.243 | 0.783 | 0.923 | 0.963 |
| | Waymo | 0.202 | 2.093 | 8.340 | 0.271 | 0.663 | 0.891 | 0.966 |
| | Foggy | 0.126 | 1.556 | 8.651 | 0.197 | 0.840 | 0.950 | 0.979 |
| | Cloudy | 0.164 | 1.829 | 8.179 | 0.232 | 0.782 | 0.929 | 0.971 |
| 0.5 | Rainy | 0.192 | 2.452 | 10.245 | 0.251 | 0.691 | 0.909 | 0.974 |
| | Sunny | 0.174 | 1.899 | 8.131 | 0.243 | 0.784 | 0.923 | 0.963 |
| | Waymo | 0.204 | 2.107 | 8.396 | 0.273 | 0.657 | 0.899 | 0.965 |
| | Foggy | 0.127 | 1.574 | 8.762 | 0.199 | 0.838 | 0.947 | 0.978 |
| | Cloudy | 0.164 | 1.826 | 8.187 | 0.232 | 0.782 | 0.929 | 0.971 |
| 0.6 | Rainy | 0.192 | 2.448 | 10.250 | 0.251 | 0.691 | 0.909 | 0.974 |
| | Sunny | 0.174 | 1.896 | 8.132 | 0.243 | 0.784 | 0.923 | 0.963 |
| | Waymo | 0.205 | 2.117 | 8.425 | 0.275 | 0.654 | 0.888 | 0.965 |
| | Foggy | 0.126 | 1.557 | 8.652 | 0.197 | 0.840 | 0.950 | 0.979 |
| | Cloudy | 0.164 | 1.830 | 8.181 | 0.232 | 0.782 | 0.929 | 0.971 |
| 0.7 | Rainy | 0.192 | 2.453 | 10.246 | 0.251 | 0.691 | 0.908 | 0.974 |
| | Sunny | 0.174 | 1.898 | 8.129 | 0.243 | 0.784 | 0.923 | 0.963 |
| | Waymo | 0.206 | 2.122 | 8.440 | 0.276 | 0.652 | 0.887 | 0.964 |
| | Foggy | 0.126 | 1.561 | 8.693 | 0.197 | 0.839 | 0.949 | 0.978 |
| | Cloudy | 0.164 | 1.825 | 8.192 | 0.232 | 0.782 | 0.929 | 0.971 |
| 0.8 | Rainy | 0.192 | 2.446 | 10.249 | 0.251 | 0.690 | 0.909 | 0.974 |
| | Sunny | 0.174 | 1.893 | 8.130 | 0.243 | 0.785 | 0.923 | 0.963 |
| | Waymo | 0.207 | 2.131 | 8.463 | 0.277 | 0.649 | 0.886 | 0.964 |
| | Foggy | 0.127 | 1.566 | 8.725 | 0.198 | 0.838 | 0.948 | 0.978 |
| | Cloudy | 0.164 | 1.826 | 8.206 | 0.233 | 0.782 | 0.928 | 0.971 |
| 0.9 | Rainy | 0.191 | 2.442 | 10.251 | 0.251 | 0.690 | 0.909 | 0.974 |
| | Sunny | 0.174 | 1.888 | 8.130 | 0.243 | 0.785 | 0.923 | 0.963 |
| | Waymo | 0.208 | 2.136 | 8.480 | 0.278 | 0.647 | 0.885 | 0.964 |
| | Foggy | 0.127 | 1.574 | 8.762 | 0.199 | 0.838 | 0.947 | 0.978 |
| | Cloudy | 0.164 | 1.824 | 8.213 | 0.233 | 0.782 | 0.928 | 0.971 |
| 1.0 | Rainy | 0.191 | 2.438 | 10.254 | 0.251 | 0.691 | 0.909 | 0.974 |
| | Sunny | 0.174 | 1.887 | 8.129 | 0.243 | 0.785 | 0.924 | 0.963 |
| | Waymo | 0.208 | 2.140 | 8.491 | 0.278 | 0.647 | 0.885 | 0.964 |
| | Foggy | 0.127 | 1.591 | 8.868 | 0.201 | 0.836 | 0.946 | 0.977 |
| | Cloudy | 0.164 | 1.821 | 8.244 | 0.234 | 0.782 | 0.928 | 0.971 |
| ∞ | Rainy | 0.191 | 2.442 | 10.306 | 0.252 | 0.689 | 0.909 | 0.974 |
| | Sunny | 0.173 | 1.878 | 8.141 | 0.243 | 0.786 | 0.924 | 0.963 |
| | Waymo | 0.217 | 2.219 | 8.637 | 0.288 | 0.627 | 0.877 | 0.960 |

Table 18: Ablation study on the number of adapted parameters in the MDE network of MonoDepth2 for adaptations to DrivingStereo and Waymo datasets. "CNN $a\%$ + BN $b\%$" indicates that only the parameters of last $a\%$ of CNN layers and last $b\%$ of BN layers in the encoder of the MDE network are adapted. Also, "Foggy", "Cloudy", "Rainy", "Sunny" indicate test cases for the DrivingStereo dataset under foggy, cloudy, rainy, and sunny weather conditions, respectively. Other conventions are the same as in Table 1 of the main text.

| Fraction | Dataset | AbsRel ↓ | SqRel ↓ | RMSE ↓ | RMSElog ↓ | $\delta < 1.25$ ↑ | $\delta < 1.25^2$ ↑ | $\delta < 1.25^3$ ↑ |
|---|---|---|---|---|---|---|---|---|
| BN 50% | Foggy | 0.149 | 2.142 | 11.000 | 0.241 | 0.779 | 0.913 | 0.964 |
| | Cloudy | 0.161 | 1.764 | 8.290 | 0.228 | 0.783 | 0.933 | 0.975 |
| | Rainy | 0.203 | 2.718 | 11.406 | 0.273 | 0.649 | 0.881 | 0.965 |
| | Sunny | 0.174 | 1.827 | 8.087 | 0.240 | 0.785 | 0.925 | 0.965 |
| | Waymo | 0.214 | 2.171 | 8.508 | 0.282 | 0.636 | 0.882 | 0.963 |
| BN 100% | Foggy | 0.127 | 1.579 | 8.600 | 0.195 | 0.840 | 0.951 | 0.980 |
| | Cloudy | 0.166 | 1.866 | 8.148 | 0.232 | 0.781 | 0.929 | 0.972 |
| | Rainy | 0.195 | 2.532 | 10.315 | 0.254 | 0.685 | 0.905 | 0.973 |
| | Sunny | 0.177 | 1.938 | 8.138 | 0.244 | 0.782 | 0.923 | 0.963 |
| | Waymo | 0.199 | 2.097 | 8.206 | 0.266 | 0.670 | 0.896 | 0.967 |
| CNN 20% + BN 80% | Foggy | 0.167 | 2.203 | 10.051 | 0.233 | 0.755 | 0.922 | 0.972 |
| | Cloudy | 0.202 | 2.311 | 9.831 | 0.275 | 0.690 | 0.899 | 0.959 |
| | Rainy | 0.208 | 2.770 | 10.875 | 0.271 | 0.644 | 0.877 | 0.965 |
| | Sunny | 0.213 | 2.445 | 9.796 | 0.283 | 0.703 | 0.895 | 0.954 |
| | Waymo | 0.828 | 18.357 | 17.197 | 0.777 | 0.241 | 0.422 | 0.572 |
| CNN 40% + BN 60% | Foggy | 0.170 | 2.299 | 10.636 | 0.243 | 0.743 | 0.911 | 0.968 |
| | Cloudy | 0.206 | 2.556 | 10.467 | 0.285 | 0.688 | 0.893 | 0.954 |
| | Rainy | 0.226 | 3.218 | 11.715 | 0.291 | 0.607 | 0.849 | 0.953 |
| | Sunny | 0.222 | 2.626 | 10.433 | 0.299 | 0.680 | 0.882 | 0.946 |
| | Waymo | 0.602 | 10.211 | 14.746 | 0.624 | 0.309 | 0.521 | 0.683 |
| CNN 60% + BN 40% | Foggy | 0.168 | 2.301 | 10.899 | 0.246 | 0.745 | 0.908 | 0.967 |
| | Cloudy | 0.206 | 2.526 | 10.311 | 0.283 | 0.690 | 0.895 | 0.955 |
| | Rainy | 0.236 | 3.559 | 12.512 | 0.306 | 0.586 | 0.833 | 0.945 |
| | Sunny | 0.226 | 2.687 | 10.476 | 0.303 | 0.672 | 0.879 | 0.945 |
| | Waymo | 0.607 | 11.791 | 15.744 | 0.638 | 0.307 | 0.526 | 0.689 |

Table 19: Ablation study on the number of adapted parameters in the MDE network of SGDepth for adaptations to DrivingStereo and Waymo datasets. "CNN $a$% + BN $b$%" indicates that only the parameters of last $a$% of CNN layers and last $b$% of BN layers in the encoder of the MDE network are adapted. Also, "Foggy", "Cloudy", "Rainy", "Sunny" indicate test cases for the DrivingStereo dataset under foggy, cloudy, rainy, and sunny weather conditions, respectively. Other conventions are the same as in Table 1 of the main text.

| Ratio | Dataset | AbsRel ↓ | SqRel ↓ | RMSE ↓ | RMSElog ↓ | $\delta < 1.25$ ↑ | $\delta < 1.25^2$ ↑ | $\delta < 1.25^3$ ↑ |
|---|---|---|---|---|---|---|---|---|
| BN 50% | Foggy | 0.127 | 1.634 | 8.787 | 0.199 | 0.837 | 0.950 | 0.979 |
| | Cloudy | 0.163 | 1.920 | 8.321 | 0.231 | 0.788 | 0.929 | 0.972 |
| | Rainy | 0.207 | 2.920 | 10.847 | 0.267 | 0.680 | 0.893 | 0.966 |
| | Sunny | 0.176 | 1.959 | 8.270 | 0.244 | 0.779 | 0.923 | 0.965 |
| | Waymo | 0.212 | 2.469 | 8.494 | 0.272 | 0.655 | 0.895 | 0.966 |
| BN 100% | Foggy | 0.127 | 1.621 | 9.023 | 0.204 | 0.830 | 0.943 | 0.975 |
| | Cloudy | 0.161 | 1.830 | 8.400 | 0.234 | 0.785 | 0.928 | 0.971 |
| | Rainy | 0.195 | 2.624 | 10.646 | 0.257 | 0.692 | 0.902 | 0.970 |
| | Sunny | 0.172 | 1.864 | 8.229 | 0.243 | 0.785 | 0.924 | 0.965 |
| | Waymo | 0.212 | 2.415 | 8.460 | 0.273 | 0.651 | 0.894 | 0.966 |
| CNN 20% + BN 80% | Foggy | 0.149 | 2.136 | 10.796 | 0.235 | 0.778 | 0.914 | 0.967 |
| | Cloudy | 0.201 | 2.299 | 10.032 | 0.282 | 0.693 | 0.895 | 0.956 |
| | Rainy | 0.222 | 3.187 | 11.436 | 0.284 | 0.627 | 0.861 | 0.955 |
| | Sunny | 0.212 | 2.448 | 10.223 | 0.294 | 0.697 | 0.884 | 0.946 |
| | Waymo | 0.363 | 5.748 | 12.352 | 0.428 | 0.448 | 0.723 | 0.869 |
| CNN 40% + BN 60% | Foggy | 0.154 | 2.179 | 10.987 | 0.238 | 0.772 | 0.912 | 0.967 |
| | Cloudy | 0.202 | 2.439 | 10.699 | 0.292 | 0.690 | 0.889 | 0.951 |
| | Rainy | 0.247 | 3.928 | 12.717 | 0.315 | 0.573 | 0.817 | 0.933 |
| | Sunny | 0.215 | 2.505 | 10.519 | 0.300 | 0.689 | 0.880 | 0.944 |
| | Waymo | 0.412 | 6.776 | 13.235 | 0.468 | 0.404 | 0.673 | 0.837 |
| CNN 60% + BN 40% | Foggy | 0.156 | 2.281 | 11.296 | 0.243 | 0.763 | 0.908 | 0.964 |
| | Cloudy | 0.206 | 2.476 | 10.764 | 0.296 | 0.684 | 0.886 | 0.950 |
| | Rainy | 0.272 | 4.875 | 14.103 | 0.343 | 0.548 | 0.788 | 0.910 |
| | Sunny | 0.219 | 2.601 | 10.776 | 0.306 | 0.680 | 0.874 | 0.940 |
| | Waymo | 0.887 | 27.394 | 19.196 | 0.763 | 0.269 | 0.473 | 0.624 |

Table 20: Ablation study on the number of adapted parameters in the MDE network of HR-Depth for adaptations to DrivingStereo and Waymo datasets. "CNN $a\%$ + BN $b\%$" indicates that only the parameters of last $a\%$ of CNN layers and last $b\%$ of BN layers in the encoder of the MDE network are adapted. Also, "Foggy", "Cloudy", "Rainy", "Sunny" indicate test cases for the DrivingStereo dataset under foggy, cloudy, rainy, and sunny weather conditions, respectively. Other conventions are the same as in Table 1 of the main text.

| Ratio | Dataset | AbsRel ↓ | SqRel ↓ | RMSE ↓ | RMSElog ↓ | $\delta < 1.25$ ↑ | $\delta < 1.25^2$ ↑ | $\delta < 1.25^3$ ↑ |
|---|---|---|---|---|---|---|---|---|
| BN 50% | Foggy | 0.127 | 1.634 | 8.787 | 0.199 | 0.837 | 0.950 | 0.979 |
| | Cloudy | 0.163 | 1.920 | 8.321 | 0.231 | 0.788 | 0.929 | 0.972 |
| | Rainy | 0.207 | 2.920 | 10.847 | 0.267 | 0.680 | 0.893 | 0.966 |
| | Sunny | 0.176 | 1.959 | 8.270 | 0.244 | 0.779 | 0.923 | 0.965 |
| | Waymo | 0.206 | 2.253 | 8.175 | 0.268 | 0.659 | 0.897 | 0.967 |
| BN 100% | Foggy | 0.129 | 1.500 | 8.656 | 0.194 | 0.828 | 0.951 | 0.983 |
| | Cloudy | 0.169 | 1.833 | 8.239 | 0.237 | 0.772 | 0.927 | 0.968 |
| | Rainy | 0.193 | 2.514 | 10.732 | 0.256 | 0.687 | 0.907 | 0.973 |
| | Sunny | 0.171 | 1.711 | 7.970 | 0.240 | 0.787 | 0.924 | 0.964 |
| | Waymo | 0.205 | 2.213 | 8.173 | 0.267 | 0.662 | 0.900 | 0.968 |
| CNN 20% + BN 80% | Foggy | 0.132 | 1.581 | 8.505 | 0.191 | 0.833 | 0.956 | 0.985 |
| | Cloudy | 0.168 | 1.907 | 7.939 | 0.230 | 0.781 | 0.931 | 0.971 |
| | Rainy | 0.209 | 2.928 | 11.090 | 0.267 | 0.672 | 0.897 | 0.969 |
| | Sunny | 0.169 | 1.699 | 7.732 | 0.233 | 0.788 | 0.928 | 0.969 |
| | Waymo | 0.584 | 11.309 | 15.746 | 0.563 | 0.300 | 0.548 | 0.740 |
| CNN 40% + BN 60% | Foggy | 0.132 | 1.588 | 8.521 | 0.191 | 0.832 | 0.955 | 0.985 |
| | Cloudy | 0.168 | 1.907 | 7.943 | 0.230 | 0.781 | 0.931 | 0.971 |
| | Rainy | 0.210 | 2.957 | 11.132 | 0.268 | 0.670 | 0.896 | 0.969 |
| | Sunny | 0.169 | 1.702 | 7.739 | 0.233 | 0.787 | 0.928 | 0.969 |
| | Waymo | 0.637 | 11.634 | 15.289 | 0.637 | 0.306 | 0.518 | 0.677 |
| CNN 60% + BN 40% | Foggy | 0.132 | 1.589 | 8.524 | 0.191 | 0.832 | 0.955 | 0.984 |
| | Cloudy | 0.206 | 2.476 | 10.764 | 0.296 | 0.684 | 0.886 | 0.950 |
| | Rainy | 0.272 | 4.875 | 14.103 | 0.343 | 0.548 | 0.788 | 0.910 |
| | Sunny | 0.219 | 2.601 | 10.776 | 0.306 | 0.680 | 0.874 | 0.940 |
| | Waymo | 0.586 | 9.640 | 14.759 | 0.612 | 0.310 | 0.523 | 0.691 |

Table 21: Ablation study on the number of adapted parameters in the MDE network of MonoViT for adaptations to DrivingStereo and Waymo datasets. "Core $a\%$ + Norm $b\%$" indicates that only the parameters of last $a\%$ of layers in the feature extractor and last $b\%$ of normalization layers in the encoder of the MDE network are adapted. Also, "Foggy", "Cloudy", "Rainy", "Sunny" indicate test cases for the DrivingStereo dataset under foggy, cloudy, rainy, and sunny weather conditions, respectively. Other conventions are the same as in Table 1 of the main text.

| Ratio | Dataset | AbsRel ↓ | SqRel ↓ | RMSE ↓ | RMSElog ↓ | $\delta < 1.25$ ↑ | $\delta < 1.25^2$ ↑ | $\delta < 1.25^3$ ↑ |
|---|---|---|---|---|---|---|---|---|
| | Foggy | 0.113 | 1.231 | 7.837 | 0.174 | 0.861 | 0.963 | 0.987 |
| | Cloudy | 0.142 | 1.568 | 7.599 | 0.204 | 0.820 | 0.945 | 0.981 |
| Norm 50% | Rainy | 0.205 | 2.585 | 10.049 | 0.258 | 0.694 | 0.900 | 0.968 |
| | Sunny | 0.149 | 1.530 | 7.587 | 0.213 | 0.814 | 0.939 | 0.977 |
| | Waymo | 0.198 | 2.160 | 7.905 | 0.256 | 0.682 | 0.908 | 0.971 |
| | Foggy | 0.117 | 1.368 | 8.699 | 0.186 | 0.844 | 0.951 | 0.984 |
| | Cloudy | 0.134 | 1.413 | 7.542 | 1.999 | 0.831 | 0.950 | 0.982 |
| Norm 100% | Rainy | 0.175 | 2.078 | 9.769 | 0.236 | 0.725 | 0.924 | 0.979 |
| | Sunny | 0.141 | 1.425 | 7.528 | 0.207 | 0.826 | 0.948 | 0.979 |
| | Waymo | 0.195 | 2.015 | 7.857 | 0.255 | 0.680 | 0.907 | 0.972 |
| | Foggy | 0.183 | 1.869 | 9.225 | 0.238 | 0.706 | 0.927 | 0.980 |
| | Cloudy | 0.186 | 1.856 | 8.408 | 0.250 | 0.715 | 0.917 | 0.972 |
| Core 20% + Norm 80% | Rainy | 0.237 | 3.208 | 11.495 | 0.295 | 0.625 | 0.867 | 0.953 |
| | Sunny | 0.185 | 1.703 | 7.960 | 0.246 | 0.742 | 0.922 | 0.968 |
| | Waymo | 0.288 | 3.101 | 9.577 | 0.348 | 0.504 | 0.797 | 0.933 |
| | Foggy | 0.304 | 6.118 | 15.728 | 0.398 | 0.527 | 0.764 | 0.883 |
| | Cloudy | 0.300 | 5.007 | 13.513 | 0.383 | 0.563 | 0.797 | 0.903 |
| Core 40% + Norm 60% | Rainy | 0.336 | 6.891 | 15.876 | 0.394 | 0.494 | 0.745 | 0.890 |
| | Sunny | 0.423 | 9.264 | 16.407 | 0.471 | 0.448 | 0.701 | 0.833 |
| | Waymo | 0.638 | 13.673 | 16.433 | 0.610 | 0.280 | 0.518 | 0.707 |
| | Foggy | 0.310 | 6.670 | 16.038 | 0.400 | 0.535 | 0.769 | 0.885 |
| | Cloudy | 0.321 | 5.232 | 14.160 | 0.423 | 0.506 | 0.768 | 0.883 |
| Core 60% + Norm 40% | Rainy | 0.324 | 6.278 | 15.382 | 0.389 | 0.500 | 0.761 | 0.895 |
| | Sunny | 0.440 | 9.718 | 16.772 | 0.487 | 0.433 | 0.685 | 0.828 |
| | Waymo | 0.585 | 14.226 | 17.371 | 0.571 | 0.322 | 0.571 | 0.749 |

Table 22: Ablation study on the number of adapted parameters in the MDE network of Lite-Mono for adaptations to DrivingStereo and Waymo datasets. "Core $a\%$ + Norm $b\%$" indicates that only the parameters of last $a\%$ of layers in the feature extractor and last $b\%$ of normalization layers in the encoder of the MDE network are adapted. Also, "Foggy", "Cloudy", "Rainy", "Sunny" indicate test cases for the DrivingStereo dataset under foggy, cloudy, rainy, and sunny weather conditions, respectively. Other conventions are the same as in Table 1 of the main text.

| Ratio | Dataset | AbsRel ↓ | SqRel ↓ | RMSE ↓ | RMSElog ↓ | $\delta < 1.25$ ↑ | $\delta < 1.25^2$ ↑ | $\delta < 1.25^3$ ↑ |
|---|---|---|---|---|---|---|---|---|
| Norm 50% | Foggy | 0.128 | 1.587 | 8.694 | 0.197 | 0.833 | 0.950 | 0.981 |
| | Cloudy | 0.158 | 1.805 | 8.260 | 0.228 | 0.789 | 0.933 | 0.972 |
| | Rainy | 0.222 | 3.167 | 11.324 | 0.282 | 0.649 | 0.878 | 0.960 |
| | Sunny | 0.173 | 1.925 | 8.190 | 0.238 | 0.781 | 0.922 | 0.967 |
| | Waymo | 0.216 | 2.644 | 8.674 | 0.277 | 0.660 | 0.894 | 0.961 |
| Norm 100% | Foggy | 0.125 | 1.509 | 8.556 | 0.194 | 0.837 | 0.951 | 0.981 |
| | Cloudy | 0.157 | 1.744 | 8.293 | 0.229 | 0.788 | 0.932 | 0.972 |
| | Rainy | 0.211 | 2.917 | 11.085 | 0.271 | 0.662 | 0.888 | 0.965 |
| | Sunny | 0.171 | 1.844 | 7.970 | 0.240 | 0.787 | 0.924 | 0.964 |
| | Waymo | 0.211 | 2.350 | 8.376 | 0.272 | 0.653 | 0.897 | 0.965 |
| Core 20% + Norm 80% | Foggy | 0.340 | 4.892 | 14.028 | 0.408 | 0.450 | 0.720 | 0.876 |
| | Cloudy | 0.377 | 5.365 | 14.042 | 0.472 | 0.420 | 0.680 | 0.834 |
| | Rainy | 0.458 | 8.537 | 15.190 | 0.481 | 0.428 | 0.682 | 0.829 |
| | Sunny | 0.340 | 3.719 | 11.001 | 0.394 | 0.484 | 0.758 | 0.902 |
| | Waymo | 0.440 | 6.539 | 12.497 | 0.473 | 0.375 | 0.652 | 0.830 |
| Core 40% + Norm 60% | Foggy | 0.346 | 4.832 | 13.461 | 0.405 | 0.438 | 0.723 | 0.883 |
| | Cloudy | 0.406 | 6.105 | 14.945 | 0.512 | 0.389 | 0.641 | 0.802 |
| | Rainy | 0.479 | 9.001 | 15.343 | 0.493 | 0.417 | 0.673 | 0.823 |
| | Sunny | 0.352 | 3.964 | 11.484 | 0.408 | 0.455 | 0.747 | 0.893 |
| | Waymo | 0.421 | 6.392 | 12.476 | 0.462 | 0.386 | 0.671 | 0.845 |
| Core 60% + Norm 40% | Foggy | 0.386 | 5.926 | 15.073 | 0.451 | 0.398 | 0.662 | 0.838 |
| | Cloudy | 0.404 | 6.217 | 15.654 | 0.530 | 0.393 | 0.636 | 0.792 |
| | Rainy | 0.503 | 9.884 | 15.845 | 0.513 | 0.407 | 0.660 | 0.810 |
| | Sunny | 0.382 | 4.759 | 12.995 | 0.452 | 0.403 | 0.687 | 0.856 |
| | Waymo | 0.530 | 9.551 | 13.991 | 0.548 | 0.334 | 0.587 | 0.767 |

