# OpenReview forum: "No Pose Estimation? No Problem: Pose-Agnostic and Instance-Aware Test-Time Adaptation for Monocular Depth Estimation"
_ICLR.cc/2026/Conference — ICLR 2026 Conference Withdrawn Submission_

### Official Review · Reviewer_ZQGT · 2025-10-30

**Soundness:** 2
**Presentation:** 3
**Contribution:** 2
**Rating:** 2
**Confidence:** 4

**Summary:**

The paper propose a method for test-time adaptation of a monocular depth estimation method. Unlike prior works, the proposed method does not require camera poses predict during test-time adaptation. The method does not assume static scene (Sfm assumotion), but rely on a pre-trained segmentation depth network to extract dynamic objects and object edges, which provide supervisory signals for test-time adaptation of a monocular depth network.

**Strengths:**

- The paper is well-written and easy to follow.
- The paper provide in-depth analysis of the failure cases of existing works, with detailed explanation for the method section.

**Weaknesses:**

Method:
- The method aims to improve depth of dynamic objects via median filtering. Such improved depth are then used  as pseudo ground-truth depth to supervise the depth network. I have several concerns regarding this. First, and most importantly, I don't get the intuition for why median filtering improve depth of dynamic objects, and thus can be used as a target of learning. Please provide clarification on this. Second, it is unclear how the method classifies between static and dynamic objects, given that no camera or object motions are predicted.
- The paper argue that camera pose might encounter the problem of domain shift if it was used during test-time adaption. However, a pre-trained segmentation network is used to obtain the object masks. This raise a question whether this segmentation model also encounter the problem of domain shift?
- Regarding the edge loss function, are the images edges and depth edges have different range (since pixel range from 0 to 1, and depth can have an arbitrary range)? It seems a bit strange to enforce images edges and depth edges being similar to each other, given their unmatched scales.

Experiments:
- It seems that the proposed method lead to an increase in the error of some models, for example, DepthAnythingV2, MonoViT, HRDepth in Tab.3. Why is that?

**Questions:**

Please refer to the questions in weaknesses section.

---

### Official Review · Reviewer_knWg · 2025-10-30

**Soundness:** 1
**Presentation:** 2
**Contribution:** 1
**Rating:** 2
**Confidence:** 4

**Summary:**

This paper proposes a novel test-time adaptation (TTA) framework for monocular depth estimation (MDE), called PITTA.
The proposed method effectively leverages instance-wise masks for dynamic objects and incorporates an additional edge-extraction–based loss function to improve adaptation performance.
Extensive experiments on the DrivingStereo and Waymo datasets demonstrate the superior performance of PITTA.

**Strengths:**

1. This method is simple and broadly applicable to most depth estimation methods.
2. Extensive evaluations on diverse benchmark datasets verify that the method achieves state-of-the-art performance.

**Weaknesses:**

## Major Weaknesses

**1. Omission of essential information:**
 - The authors did not provide qualitative comparisons between the proposed method and prior works. This evaluation is crucial, especially to demonstrate the advantages and distinctions introduced by the edge extraction component.

 - In the Overview of Architecture section, the authors allocated an excessive portion of the content to explaining related work rather than describing the proposed method.

- The implementation details of the test-time adaptation (TTA) procedure are insufficient and should be described in more depth.

**2. Concerns regarding technical novelty:**

 - The authors did not adequately explain the motivation and effect of using the median filter, nor did they discuss its implications.

 - Although the method incorporates several components from prior work, it seems to lack a sufficiently novel contribution that can be considered fundamentally original.

 - In line 259, the authors appear to oversimplify existing methodologies of self-supervised monocular depth estimation (SSMDE). The auto-masking strategy in Monodepth2 is not merely designed to handle static objects; rather, it addresses regions where the object’s motion aligns with the camera motion, making them appear stationary. Claiming that all static objects degrade training performance is an overstatement. In fact, depth can still be learned through temporal consistency constraints even for static objects. Such oversimplification weakens the authors’ argument, and additional experiments or thorough justification are required to strengthen the foundational assumptions of the paper.

 - Additionally, although the title and primary claim of the paper emphasize a "pose-agnostic" framework, the justification for this assertion is insufficient. The authors should provide a clearer explanation of how the proposed method achieves pose-agnostic behavior, supported by thorough reasoning. Furthermore, quantitative evaluations are necessary to substantiate this claim and demonstrate its robustness across varying pose conditions.

**3. Concerns regarding experimental results:**

 - In Tables 1 and 2, the results show that the “No adaptation” baseline performs better than previous TTA works. The authors should provide justification or analysis to explain this phenomenon.

 - It is unclear whether the evaluation of DepthAnythingV2 was conducted in a zero-shot manner or with TTA applied. If not adapted, the authors should discuss whether applying the proposed method could lead to meaningful improvements.

 - In the supplementary material (e.g., Tables 10, 12, and 14), applying the proposed method appears to degrade performance in some cases. This requires explanation and analysis.


--------

## Minor Weaknesses

**1. Writing quality:**

The authors cited references in a manner such as “NewCRFs Yuan et al. (2022);”, which reduces readability. It is recommended to revise the citation style with reference to papers such as [T1, T2, T3]. Additionally, typographical issues (e.g., “Appendix ?” at line 254) further detract from readability. Improving writing clarity will help present the contributions more effectively.

**2. Indicating best and second-best results in tables:**

The authors did not consistently highlight the best and second-best scores in the tables. Revising this formatting would make quantitative improvements clearer and more convincing.



--------

[T1] An Image is Worth 16x16 Words: Transformers for Image Recognition at Scale, ICLR 2021.

[T2] Score-Based Generative Modeling through Stochastic Differential Equations, ICLR 2021.

[T3] Depth Pro: Sharp Monocular Metric Depth in Less Than a Second, ICLR 2025.

**Questions:**

First of all, I would like to express my gratitude for the invaluable work of the authors, which contributes to the advancement of this field.

Here are some questions and points that require further clarification:

1. Could the authors provide a clear explanation for the rationale behind applying the median filter and offer convincing justifications for its use?

2. What specific reasons allow the proposed method to be considered pose-agnostic? Additionally, could the authors provide further experimental results or evidence to support this claim?

3. I would appreciate additional details regarding the implementation of the Test-Time Adaptation (TTA) and the evaluation methods and policies used to assess the baseline models.

4. Further clarification is needed regarding the mention of Depth Anything V2 and the “no adaptation” scenario, as highlighted in the major weaknesses section.

5. A rebuttal and explanation for the third point mentioned in the major weaknesses section, specifically in line 259, would be appreciated.

6. Finally, I would like to request a supplementary explanation for the observed trend where the performance appears to degrade when the proposed method is applied to the quantitative evaluations presented in the supplementary material (e.g., Table 10, 12, and 14).

---

### Official Review · Reviewer_uvYR · 2025-10-31

**Soundness:** 3
**Presentation:** 3
**Contribution:** 2
**Rating:** 4
**Confidence:** 3

**Summary:**

The paper proposes a pose-agnostic, instance-aware test-time adaptation (TTA) framework for monocular depth estimation (MDE). The method avoids pose supervision by assuming identity rotation and zero translation during TTA and adapts only batch-norm parameters in the encoder. Two additional components are used: (1) instance-aware masking from a frozen panoptic segmenter, and (2) an edge-guided loss.

**Strengths:**

- Clear motivation for making TTA independent of pose estimation, which is often brittle under domain shift.

- Simple and modular: BN-only updates and plug-in components (instance masks and edge guidance) make it easy to integrate with standard MDE backbones.

**Weaknesses:**

Previously, we did evaluation of DepthAnything v2 on DrivingStereo fog/rain yields **δ<1.25 of 96.3/94.9**, whereas the paper reports **73.6/65.7**. This large gap raises concerns about experimental validity. It is our major concern.

The approach critically depends on a *frozen panoptic segmenter* to create instance masks that directly modulate supervision. In adverse conditions (fog/rain/night/unusual classes), segmentation quality may degrade, potentially harming adaptation. The paper lacks a robustness analysis of mask errors.

Experiments cover fog and rain (and an “All” mix), but night and snow are also crucial deployment shifts. Please either provide the evaluations or justify their exclusion and discuss expected behavior.

**Questions:**

With R=I, t=0 during TTA, is any **temporal information** used beyond frame streaming? How does the method behave under strong inter-frame motion?

Report per-frame costs for segmentation and adaptation (hardware, iterations, wall-clock latency) relevant for on-vehicle deployment.

---

### Official Review · Reviewer_F3tC · 2025-10-31

**Soundness:** 3
**Presentation:** 3
**Contribution:** 3
**Rating:** 6
**Confidence:** 4

**Summary:**

This paper proposes a novel pose-agnostic TTA framework named PITTA for monocular depth estimation, which uses instance-aware segmentation masks to create new self-supervised losses, thereby avoiding failures caused by the unreliable pose estimation in traditional TTA methods and achieving SOTA results on two major benchmarks.

**Strengths:**

1.	Clear motivation: The paper rightly points out and illustrates that depending on unreliable pose networks is one of the main weaknesses in the current TTA techniques.

2.	Novel Compensation: The presented self-supervised losses (depth-refining and edge-guided), that essentially replace temporal consistency with semantic consistency.

3.	Extensive & SOTA Experiments: The approach is state-of-the-art on two datasets (DrivingStereo, Waymo) and is demonstrated to be a general, plug-and-play method by enhancing five MDE backbones.

**Weaknesses:**

Dependency Swap: The approach does not eliminate dependencies, it only replaces the dependence on a pose network with the dependence on an equally complicated panoptic segmentation network.

Unverified Core Assumption: The approach presumes that the frozen segmentation network is resilient to the same domain changes (e.g. fog, rain) that the MDE model encounters. This is an important assumption that is not tested or discussed.

Lacking Computational Analysis: The computer analysis (e.g., FPS) is not done anywhere in the paper. The TTA process, which necessitates a forward pass along with an MDE backward pass at each frame (both potentially time-intensive), is not potentially fast enough to be used in practice in a real-time application.

**Questions:**

(W1) Methodologically, why is a dependency on segmentation fundamentally better than a dependency on pose estimation?

(W2) How robust is the frozen Mask2Former to the same adverse weather? What happens to PITTA's performance when segmentation quality degrades?

(W3) What is the total computational overhead (e.g., FPS or ms/frame) of PITTA, including the Mask2Former pass and the MDE backward pass?

---

### Note · Authors · 2025-11-14

I have read and agree with the venue's withdrawal policy on behalf of myself and my co-authors.